# ZFP36 RNA-binding proteins restrain T cell activation and anti-viral immunity

Michael J Moore[1†], Nathalie E Blachere[1], John J Fak[1], Christopher Y Park[1,2], Kirsty Sawicka[1], Salina Parveen[1], Ilana Zucker-Scharff[1], Bruno Moltedo[3], Alexander Y Rudensky[3], Robert B Darnell[1,2]*

[1]Laboratory of Molecular Neuro-Oncology, Howard Hughes Medical Institute, The Rockefeller University, New York, United States; [2]New York Genome Center, New York, United States; [3]Howard Hughes Medical Institute, Ludwig Center at Memorial Sloan Kettering Cancer Center, New York, United States

**Abstract** Dynamic post-transcriptional control of RNA expression by RNA-binding proteins (RBPs) is critical during immune response. ZFP36 RBPs are prominent inflammatory regulators linked to autoimmunity and cancer, but functions in adaptive immunity are less clear. We used HITS-CLIP to define ZFP36 targets in mouse T cells, revealing unanticipated actions in regulating T-cell activation, proliferation, and effector functions. Transcriptome and ribosome profiling showed that ZFP36 represses mRNA target abundance and translation, notably through novel AU-rich sites in coding sequence. Functional studies revealed that ZFP36 regulates early T-cell activation kinetics cell autonomously, by attenuating activation marker expression, limiting T cell expansion, and promoting apoptosis. Strikingly, loss of ZFP36 in vivo accelerated T cell responses to acute viral infection and enhanced anti-viral immunity. These findings uncover a critical role for ZFP36 RBPs in restraining T cell expansion and effector functions, and suggest ZFP36 inhibition as a strategy to enhance immune-based therapies.
DOI: https://doi.org/10.7554/eLife.33057.001

*For correspondence:
darnelr@rockefeller.edu

Present address: †Regeneron Pharmaceuticals, Tarrytown, United States

## Introduction

Immune responses require precise, dynamic gene regulation that must activate rapidly as threats rise and resolve efficiently as they clear. Post-transcriptional control of mRNA abundance and expression by RNA binding proteins (RBPs) is a key layer of this response that can enact rapid, signal-responsive changes (*Kafasla et al., 2014*; *Hao and Baltimore, 2009*), but knowledge of specific functional roles for dedicated RBPs remains limited.

AU-rich elements (AREs) in mRNA 3'-untranslated regions (3'-UTR) facilitate post-transcriptional control of many immune functions, including cytokine expression, signal transduction, and immediate-early transcriptional response (*Chen and Shyu, 1994*; *Shaw and Kamen, 1986*; *Caput et al., 1986*). Many RBPs bind AREs, with diverse ensuing effects on RNA turnover, translation, and localization (*Stoecklin and Mühlemann, 2013*; *Tiedje et al., 2012*; *von Roretz et al., 2011*). The Zinc finger binding protein 36 (ZFP36) family of proteins are prototypical ARE-binding factors with distinctive, activation-dependent expression in hematopoietic cell lineages (*Raghavan et al., 2001*; *Carballo et al., 1998*). The family includes three somatic paralogs: ZFP36 (a.k.a Tristetraprolin, TTP), ZFP36L1 (a.k.a butyrate responsive factor 1, BRF1), and ZFP36L2 (a.k.a BRF2) with highly homologous CCCH zinc-finger RNA binding domains (*Blackshear, 2002*).

In many contexts, the archetypal paralog ZFP36 de-stabilizes target mRNAs by binding 3'-UTR AREs and recruiting deadenylation and degradation factors (*Brooks and Blackshear, 2013*; *Lykke-Andersen and Wagner, 2005*). More recently, evidence has begun to emerge for roles in translation (*Tao and Gao, 2015*; *Tiedje et al., 2012*) but in vivo function and context have not been

**eLife digest** The immune system must quickly respond to anything that may cause disease – from cancerous cells to viruses. For instance, a type of white blood cell called a T cell patrols the body, looking for potential threats. If a T cell identifies such a threat, it "activates" and undergoes various changes so that it can help to eliminate the problem.

One way that T cells change is by switching on different genes to make specific proteins. The information in the genes is first used as a template to produce a molecule called a messenger RNA (mRNA), which is then translated to build proteins. So-called RNA-binding proteins help control events before, during and after the translation stage in the process. Previous studies have shown that one particular RNA-binding protein, called ZFP36, controls the translation of proteins that are important for how the immune system recognizes the body's own tissue and deals with cancer cells. However, it was less clear if it also helped T cells to activate and defeat viruses.

Now, using cutting-edge technology, Moore et al. have identified thousands of new mRNAs controlled by ZFP36 in mice, many of which did indeed make proteins that help T cells activate and spread throughout the body. Further experiments showed that mice that lack ZFP36 in the T cells were much quicker at responding to viruses than other mice. This suggests that ZFP36 actually restrains T cells and slows down the body's immune system.

Knowing more about how T cells work could lead to new treatments for diseases; it may, for example, allow scientists to engineer T cells to better attack cancer cells, However, other studies have shown that mice without ZFP36 often go on to develop autoimmune diseases, which result from the immune system attacking healthy cells by mistake. As such, it seems that there is a fine line between improving the body's immune system and increasing the risk of autoimmune diseases, and that RNA-binding proteins play an important role in managing this delicate balance.
DOI: https://doi.org/10.7554/eLife.33057.002

established. While many aspects remain unsettled, ZFP36 is clearly critical for immune function, as its loss causes a systemic inflammatory disease in mice (*Taylor et al., 1996*). A key feature of this syndrome is aberrant stabilization and over-expression of *Tnf* in myeloid cells, particularly macrophages (*Carballo et al., 1998*), where UV cross-linking and immunoprecipitation (CLIP) analyses have further supported direct regulation (*Tiedje et al., 2016*; *Sedlyarov et al., 2016*). However, this role does not fully account for ZFP36 function in vivo, as underscored by reports that myeloid-specific deletions of *Zfp36* do not recapitulate spontaneous autoimmunity (*Qiu et al., 2012*; *Kratochvill et al., 2011*).

Increasing evidence points to important functions for ZFP36 proteins in adaptive immunity. Dual ablation of paralogs *Zfp36l1* and *Zfp36l2* in T cells arrests thymopoeisis at the double-negative stage, and causes lethal lymphoma linked to *Notch1* dysregulation (*Hodson et al., 2010*). This role in restraining aberrant proliferation was later extended to B-cell development and lymphoma (*Galloway et al., 2016*; *Rounbehler et al., 2012*), but the severe phenotype precluded analysis of ZFP36 family function in mature T cells. Consistent with such a function, in vitro studies suggest ZPF36 regulates the expression of T cell-derived cytokines, including IL-2, IFN-γ, and IL-17, that mediate lymphocyte homeostasis, microbial response, and inflammation (*Lee et al., 2012*; *Ogilvie et al., 2009*; *2005*). The landscape of ZFP36 targets beyond these limited cases in T cells is unknown, but will be the key to understanding its emerging roles in inflammation, autoimmunity, and malignant cell growth (*Patial and Blackshear, 2016*).

To determine ZFP36 functions in T cells, we employed high-throughput sequencing of UV-cross-linking and immunoprecipitation (HITS-CLIP) to generate a definitive set of ZFP36 RNA targets. HITS-CLIP utilizes in vivo UV-cross-linking to induce covalent bonds between RBPs and target RNAs, allowing stringent immunopurification and thus rigorous identification of direct binding events (*Licatalosi et al., 2008*; *Ule et al., 2003*). These new ZFP36 RNA binding maps pointed to roles in regulating T-cell activation kinetics and proliferation, a function confirmed in extensive functional assays, and in vivo studies demonstrating a critical role in anti-viral immunity. Our results illuminate novel functions for ZFP36 in adaptive immunity, laying groundwork for understanding and modulating its activity in disease.

## Results

### ZFP36 dynamics during T-cell activation

ZFP36 expression is induced upon T-cell activation (*Raghavan et al., 2001*). We examined its precise kinetics following activation of primary mouse CD4 +T cells by Western analysis with custom ZFP36 antisera generated against a C-terminal peptide of mouse ZFP36. Protein levels peaked ~4 hr post-activation and tapered gradually through 72 hr, and were re-induced by re-stimulation 3 days post-activation (*Figure 1A*). ZFP36 expression depended on both TCR stimulation, provided by anti-CD3, and co-stimulation, provided by co-cultured dendritic cells (DCs) (*Figure 1B*). A similar pattern of transient ZFP36 induction occurred in activated CD8 +T cells (*Figure 1—figure supplement 1A*).

Western blot analysis showed multiple bands at ~40–50 kD, indicating several isoforms. Notably, isoforms running above the predicted molecular weight (MW) of ZFP36 (36 kD) pre-dominated early after activation, and are consistent with previously reported hyperphosphorylation (*Qiu et al., 2012*). In addition, partial conservation of the immunizing peptide in ZFP36L1 and ZFP36L2 raised the possibility of paralog cross-reactivity. Western analyses with recombinant constructs confirmed ZFP36, ZFP36L1, and ZFP36L2 are readily detected with our custom antisera (henceforth, pan anti-ZFP36; *Figure 1—figure supplement 1B*). Commercial paralog-specific antibodies were identified, and Western analysis showed that both ZFP36 and ZFP36L1 were induced by T-cell activation (*Figure 1—figure supplement 1B–C*). ZFP36L2, expected to run at ~62 kD, was not detected under any conditions examined. Analysis of *Zfp36* KO T cell lysates with pan ZFP36 antisera showed ~50% reduced signal compared to WT. We conclude that the residual signal is likely due to persistent expression of ZFP36L1, which is highly homologous and of similar size to ZFP36. Collectively, these results demonstrate activation-dependent expression of ZFP36 and ZFP36L1 in T cells, and suggest activated *Zfp36* KO T cells have partial loss (~50%) of pan ZFP36 expression (*Figure 1—figure supplement 1D*).

The characterization of *Zfp36* as an immediate early response gene in various cell types established transcription as a mechanism of its activation-induced expression (*Lai et al., 1995*). Interestingly, *Zfp36* (RPKM = 22.5) and *Zfp36l1* (RPKM = 8.2) mRNAs are robustly present in RNAseq data from naïve CD4 +T cells, despite an absence of detectable protein by western blotting. ZFP36L2 was not detected in any conditions examined, but its mRNA was also detected in naïve cells (RPKM = 30.2). These observations indicate that post-transcriptional mechanisms regulate expression of ZFP36 paralogs in T cells.

### Transcriptome-wide identification of ZFP36 target RNAs in CD4 +T cells

The striking pattern of ZFP36/L1 expression in T cells led us to develop ZFP36 HITS-CLIP as a screen for its biological function (*Figure 1C–F*; *Figure 1—figure supplement 2*). Notably, ZFP36/L1 RNPs isolated by CLIP from activated CD4 +T cells sera exhibited high molecular weight (MW) complexes resistant to detergent, heat, and RNAse, consistent with a pattern previously observed in ZFP36 CLIP in macrophages (*Figure 1D*, *Figure 1—figure supplement 2A*; [*Sedlyarov et al., 2016*]). This RNP signal pattern was dependent on UV irradiation, and was observed with two different anti-pan-ZFP36 but neither pre-immune sera.

Given prior evidence that ZFP36 regulates T-helper type-1 (Th1) cytokines (e.g. TNF-α and IFN-γ), we next generated a comprehensive map of ZFP36/L1 binding sites by HITS-CLIP using anti-pan-ZFP36 in WT CD4 +T cells, activated for 4 hr under Th1-polarizing conditions (*Ogilvie et al., 2009*; *Carballo et al., 1998*). 5132 robust binding sites were defined, requiring a peak height (PH) >5, and support from at least 3 (of five total) biological replicates and two different pan-ZFP36 antisera (*Supplementary file 1A*; [*Shah et al., 2017*; *Moore et al., 2014*]). Consistent with identification of bonafide ZFP36/L1 binding events, HITS-CLIP recovered the known AU-rich ZFP36 consensus motif at high significance, along with reported binding sites in *Tnf*, *Ifng* and other targets (*Figure 1E*; *Supplementary file 1A*; [*Brewer et al., 2004*]). Globally, ZFP36/L1 sites confirmed a preponderance of 3'-UTR binding (>75%), and showed substantial binding in coding sequence (CDS;~6.5%) and introns (5.4%) (*Figure 1F*). Separate analysis of low and high MW RNP complexes showed similar transcript localization and motif enrichment, all consistent with ZFP36 binding (*Figure 1—figure supplement 2B*). This analysis indicates the presence of large, stable ZFP36 complexes in vivo,

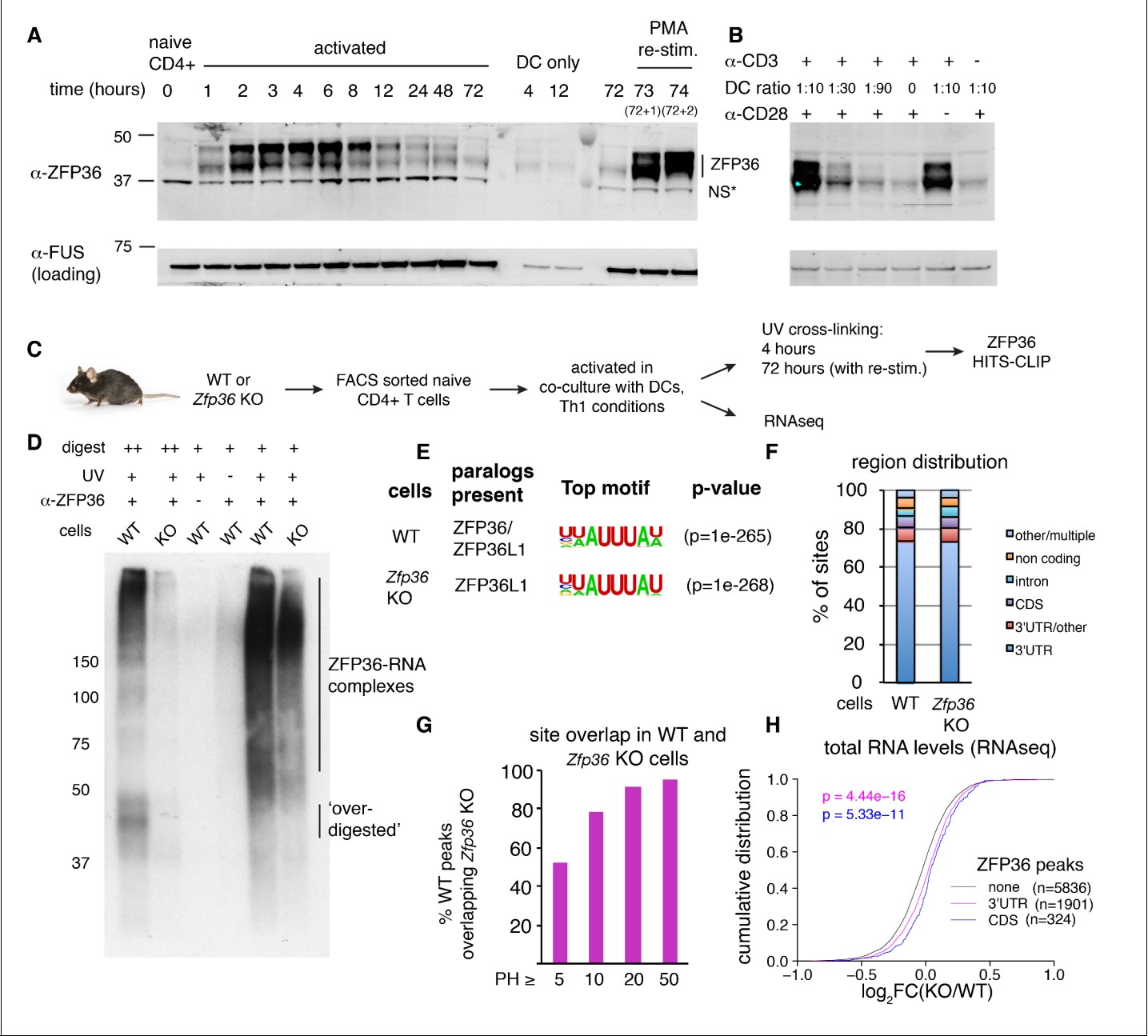

**Figure 1.** HITS-CLIP as a transcriptome-wide screen for ZFP36 function in T cells. (A) Immunoblots with pan-ZFP36 antisera after activation of naïve CD4 +T cells in DC co-cultures, and with re-stimulation at day 3. Antibody and MW markers are shown on the left. NS* indicates a non-specific band. (B) Immunoblotting with pan-ZFP36 antisera 4 hr after activation of naïve CD4 +T cells, testing dependence on TCR stimulation (α-CD3), and co-stimulation (DCs or α-CD28). (C) ZFP36 HITS-CLIP design. (D) Representative autoradiogram of ZFP36 CLIP from activated CD4 +T cells using pan-ZFP36 antisera, with pre-immune and no-UV controls. Signal in *Zfp36* KO cells is due to capture of ZFP36L1 RNP complexes. (E) The most enriched binding motifs and (F) annotation of binding sites from WT and *Zfp36* KO cells. (G) Overlap of binding sites in WT and *Zfp36* KO cells, stratified by peak height (PH). CLIP data are compilation of 4 experiments, with 3–5 total biological replicates were condition. (H) RNAseq in WT and *Zfp36* KO CD4 +T cells activated under Th1 conditions for 4 hr. Log2-transformed fold-changes (KO/WT) are plotted as a cumulative distribution function (CDF), for mRNAs with 3′UTR, CDS, or no significant ZFP36 HITS-CLIP sites. Numbers of mRNAs in each category (n) and p-values from two-tailed Kolmogorov-Smirnov (KS) tests are shown. RNAseq data is a compilation of 2 experiments, with 3–4 biological replicates per condition.

DOI: https://doi.org/10.7554/eLife.33057.003

The following figure supplements are available for figure 1:

**Figure supplement 1.** ZFP36 paralog expression in T cells.

DOI: https://doi.org/10.7554/eLife.33057.004

*Figure 1 continued on next page*

*Figure 1 continued*

**Figure supplement 2.** ZFP36 HITS-CLIP.
DOI: https://doi.org/10.7554/eLife.33057.005
**Figure supplement 3.** Regulation of mRNA abundance by ZFP36.
DOI: https://doi.org/10.7554/eLife.33057.006

consistent with stable multimers (*Cao et al., 2003*; *2004*). Subsequently, CLIP reads from different MW regions were pooled to maximize dataset depth.

To examine possible paralog specificity, we also mapped ZFP36L1 sites by HITS-CLIP in *Zfp36* KO CD4+T cells under identical conditions. As in western analysis (*Figure 1—figure supplement 1C–D*), *Zfp36* KO samples showed reduced but significant CLIP signal compared to WT (*Figure 1D*, *Figure 1—figure supplement 2A*), representing ZFP36L1-RNA complexes. Sites in WT and *Zfp36* KO cells showed very similar enriched motifs and transcript localizations, indicating that ZFP36 and ZFP36L1 have similar binding profiles in vivo (*Figure 1E–F*, *Figure 1—figure supplement 2C*, *Supplementary file 1B*). Majorities of robust sites (53%) and target mRNAs (66%) identified in WT cells were found independently in *Zfp36* KO cells, and site overlap was far greater (>90%) for peaks of increasing magnitude (*Figure 1G*). A subset of potential ZFP36L1-specific sites was identified only in *Zfp36* KO cells (n = 675; *Supplementary file 1C*), although these showed similar features to ZFP36 sites overall (*Figure 1—figure supplement 2C*; third panel). Thus, these analyses do not exclude paralog specificity at some sites, but indicate broadly overlapping in vivo RNA binding for ZFP36 and ZFP36L1 reflecting their high homology.

Secondary enriched motifs revealed additional properties of ZFP36/L1 target sites. The second top motif resembled the known recognition sequence for polyadenylation factor CFI(m)25 (*Venkataraman et al., 2005*). Accordingly, ZFP36 binding in 3'UTRs was most concentrated in the vicinity of expected polyA sites,~50 nucleotides before transcript ends (*Figure 1—figure supplement 2D*). Analysis of CDS-specific binding revealed the AU-rich ZFP36 motif, along with strong enrichment of the 5' splice site (5'-ss) consensus (*Figure 1—figure supplement 2E*). Cross-link-induced truncations (CITS) clarified that CDS peaks are centered within coding exons, but supporting CLIP reads often spanned the exon-intron boundary. Thus, at least a subset of CDS binding by ZFP36 occurs prior to pre-mRNA splicing in the nucleus.

## ZFP36 represses target mRNA abundance and translation during T-cell activation

We next employed RNA profiling strategies to determine the functional effects of ZFP36/L1 binding. RNAseq analysis in WT and *Zfp36* KO CD4 +T cells activated under conditions identical to our HITS-CLIP analyses uncovered two main effects. First, a small number of mRNAs that were silent in WT cells, including numerous immunoglobulin loci, were detected at low to moderate levels in KO cells (*Figure 1—figure supplement 3A*). These mRNAs lacked evidence of ZFP36/L1 binding, both in HITS-CLIP data and direct motifs searches, suggesting they are not direct targets. Given established chromatin regulation of many of these loci (i.e. Ig genes) and the on-off nature of the changes, dysregulated transcriptional silencing is a potential explanation, but is likely to be a secondary effect of ZFP36 loss of function. The second main effect emerged from a global analysis, which showed ZFP36 binding in 3'UTR (p=4.44×10$^{-16}$; Kolmogorov-Smirnoff [KS]) and CDS (p=5.33×10$^{-11}$) correlated to subtle but highly significant shifts toward greater mRNA abundance in *Zfp36* KO cells, relative to mRNAs with no binding sites (*Figure 1H*). This correlation was not observed for mRNAs with binding exclusively in introns, and we did not find evidence in these data that ZFP36/L1 binding correlated with altered usage of proximal splice or polyA sites (not shown). The same pattern was also observed when considering a more stringently defined of sites set overlapping statistically robust CITS (*Figure 1—figure supplement 3B*).

The overall trend in transcriptome profiling is consistent with evidence that ZFP36 represses RNA abundance (*Lykke-Andersen and Wagner, 2005*). However, stratification of sites by the magnitude of ZFP36 CLIP binding allowed resolution of potentially complex effects. For 3'UTR binding, ZFP36 targets overall showed a significant shift in abundance, but mRNAs containing the top 20% most robust sites (ranked by peak height [PH], see Materials and methods) showed no significant effect. Thus, a higher degree of binding correlated with less effect on RNA abundance in the absence of

ZFP36 (*Figure 1—figure supplement 3C*). This trend was not observed for CDS sites, where the top 20% showed a similar shift to sites overall (*Figure 1—figure supplement 3D*). Thus, our analyses show a trend of negative regulation of RNA abundance in this context, but with blunted effects for highly robust binding sites in 3'UTR (see Discussion).

We next examined in more detail the effects of ZFP36 regulation for HITS-CLIP targets with highly robust 3'UTR binding in T cells. Activation marker CD69, apoptosis regulator BCL2, and effector cytokines TNF and IFNG showed significantly increased protein levels in *Zfp36* KO versus WT T cells 4 hr after activation (*Figure 2A*, *Figure 2—figure supplement 1*). Of these, only *Bcl2* showed increased mRNA abundance. *Tnf, Ifng*, and *Cd69* were all among the top 20% of targets as defined by CLIP binding magnitude (PH), thus supporting the trend in our global analyses that some highly robust binding targets show little regulation at the level of mRNA abundance in this context. The effects on protein level in the absence of changes in mRNA abundance suggested regulation of translation. We tested this principle by constructing GFP fluorescent reporters with an intact 3'UTRs (WT-UTR), or variants with the CLIP-defined ZFP36 binding site deleted (Δ-UTR; *Figure 2B*), for *Cd69, Tnf, and Ifng*. In 293 cells, ZFP36 strongly repressed protein expression for all three WT-UTR reporters, while showing weaker (*Tnf* and *Ifng*) or no (*Cd69*) repression of mRNA levels. Of note, the Δ-UTR constructs showed increased protein levels both in the presence and absence of ZFP36. This indicates that additional, endogenous factors are regulating these AU-rich sites in 293 cells, though Western analyses confirmed that ZFP36 paralogs are undetectable (*Figure 1—figure supplement 1A*). In addition, ZFP36 over-expression exerted ~2 fold repression of Δ-UTR constructs, which may indicate incomplete ablation of binding or secondary effects of ZFP36 over-expression. However, repression of WT-UTR constructs was consistently greater than for Δ-UTR variants, demonstrating specific ZFP36 repression of the defined binding sites. In summary, these heterologous assays independently confirmed ZFP36 regulation of CLIP-defined targets, and support specific effects of translation, in addition to RNA stability.

To directly test a role for ZFP36 in translational regulation in T cells, we next performed ribosome profiling of WT and *Zfp36* KO CD4 +T cells activated for 4 hr under Th1 conditions (*Figure 3A* and *Figure 3—figure supplement 1A*; [*Ingolia et al., 2009*]). We observed robust ribosome association of *Zfp36* mRNA in WT cells that was lost completely downstream of the engineered gene disruption in *Zfp36* KO cells (*Figure 3—figure supplement 1B*), consistent with accurate identification of translating mRNAs. Globally, there was a subtle but significant shift toward greater ribosome association in *Zfp36* KO cells for mRNAs bound by ZFP36 in 3'UTR ($p=1.04\times10^{-11}$; KS) or CDS ($p=1.58\times10^{-11}$), relative to mRNAs with no ZFP36 binding (*Figure 3B*). These shifts mirror those for global RNA abundance, with two notable exceptions. First, mRNAs with ZFP36 binding in CDS showed a significantly larger shift in ribosome association than those with 3'UTR binding (*Figure 3B*). Second, in contrast to effects on RNA abundance, the top 20% most robust ZFP36 binding sites in 3'UTR showed similar effects on ribosome association to sites overall (*Figure 3—figure supplement 1C*). Thus, ZFP36 target mRNAs show increased ribosome association in *Zfp36* KO cells.

Levels of ribosome-associated mRNA are related to total abundance. To evaluate changes in translational efficiency (ΔTE) in *Zfp36* KO versus WT T cells, ribosome profiling fold-changes were normalized to those from RNAseq. We then used Gene Set Enrichment Analysis (GSEA) to examine the distribution of ZFP36 targets among all detected mRNAs ranked by ΔTE (*Subramanian et al., 2005*). Importantly, this analysis compares the observed ΔTE of CLIP-defined ZFP36 target mRNAs to mRNAs with no detected ZFP36 binding sites. ZFP36 3'UTR and, more significantly, CDS binding targets were strongly enriched for increased TE in *Zfp36* KO cells (*Figure 3C*). In addition, ZFP36 targets with highly robust 3'UTR binding showed more significant effects on TE than ZFP36 3'UTR targets overall. This enrichment was not observed for mRNAs with intronic binding sites, indicating specificity for 3'UTR and CDS binding. As a striking confirmation of these results, normalized ribosome coverage on robust 3'UTR targets *Tnf* and *Ifng* was significantly higher in *Zfp36* KO cells than WT (*Figure 3D*), despite no detectable difference in overall mRNA abundance (*Figure 2A*). Crucially, ribosome coverage averaged across all mRNAs was not appreciably different between KO and WT cells, indicating specific effects on ZFP36 targets (*Figure 3E*). Notably, the pattern of ribosome association along these and other transcripts is remarkably consistent between WT and *Zfp36 KO* cells, but with altered magnitude. Mechanistically, this observation indicates that ZFP36 prevents association of mRNAs with ribosomes, but does not impact elongation. These results indicate repression of mRNA target translation by ZFP36 during T- cell activation, likely at the level of initiation.

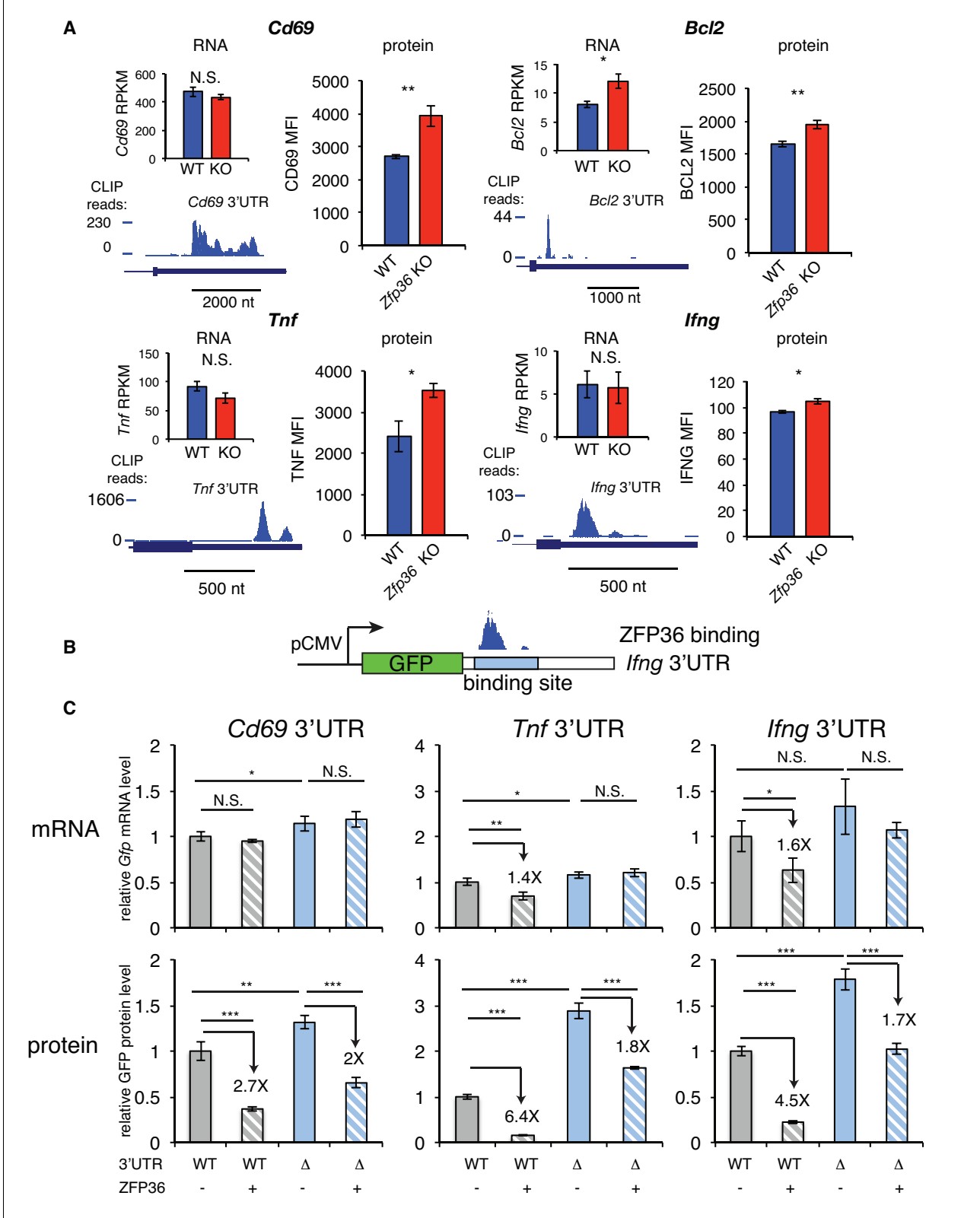

**Figure 2.** ZFP36 regulates target protein levels in T cells. (A) Levels of mRNA and protein in *Zfp36* KO and WT T cells and ZFP36 CLIP tracks measured 4 hr post-activation for targets with robust 3'UTR ZFP36 binding. RNA values are mean RPKM ± S.E.M. of 4 biological replicates. Protein values are mean fluorescence intensities (MFI) ± S.E.M. for 3–4 mice per condition. (B) Design of GFP reporters with WT 3'UTR (WT-UTR) or one lacking the ZFP36 binding site (Δ-UTR). (C) WT-UTR or Δ-UTR reporters were co-transfected into 293 cells with *Zfp36* (+) or vector alone (-). 24 hr post-transfection,

*Figure 2 continued*

reporter mRNA and protein levels were measured by RT-qPCR and flow cytometry, respectively. Values are mean ± S.D. of 4 biological replicates in each condition. Data for *Ifng* reporters show one representative experiment of three performed. *Tnf* and *Cd69* reporters were analyzed in one experiment. Results of two-tailed t-tests: *=p < 0.05; **=p < 0.01; ***=p < 0.001.

DOI: https://doi.org/10.7554/eLife.33057.007

The following figure supplement is available for figure 2:

**Figure supplement 1.** Protein levels of ZFP36 CLIP targets.

DOI: https://doi.org/10.7554/eLife.33057.008

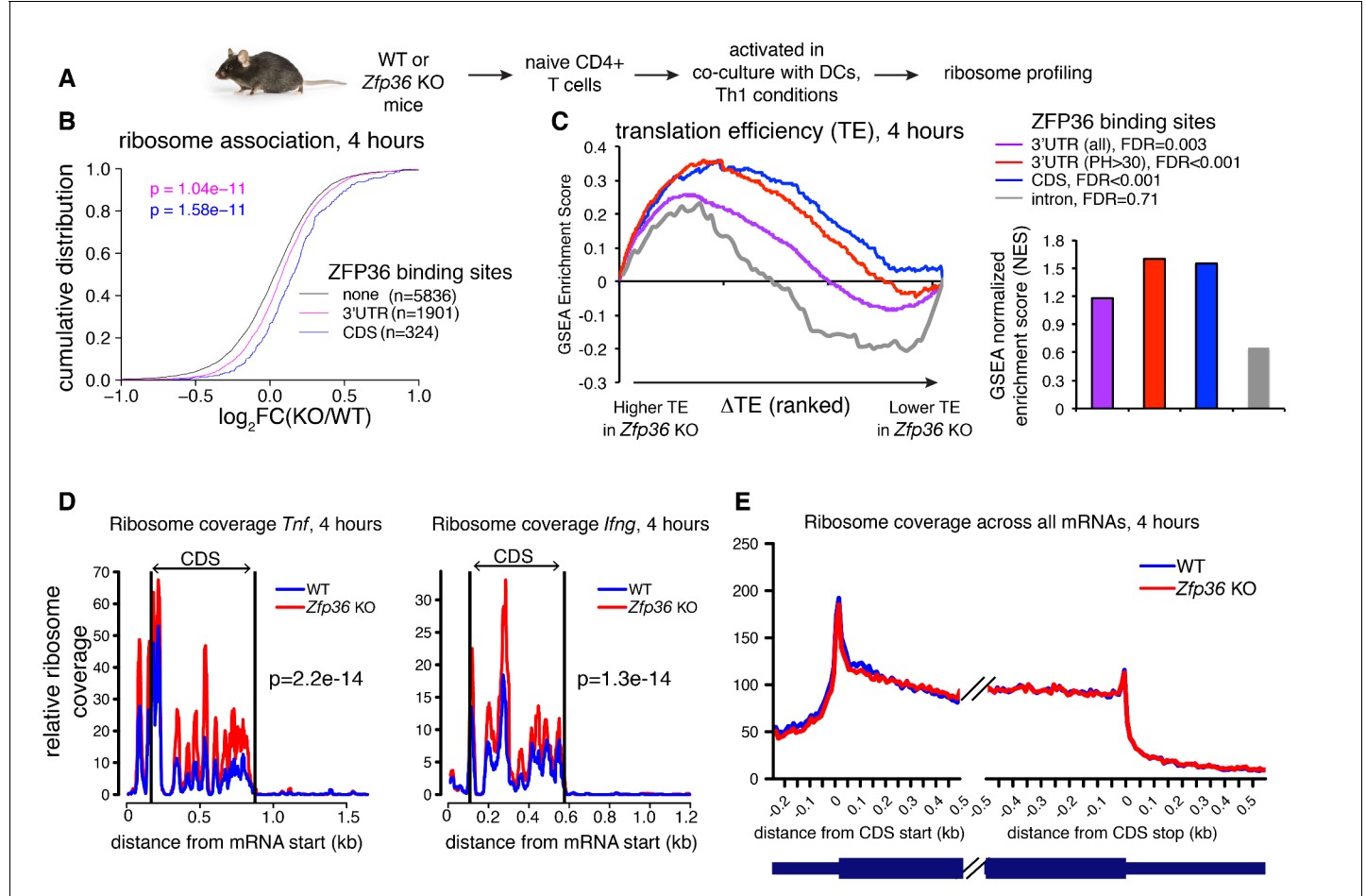

**Figure 3.** ZFP36 regulates target ribosome association. (**A**) Ribosome profiling of *Zfp36* KO and WT CD4 +T cells. (**B**) Changes in ribosome association between *Zfp36* KO and WT cells plotted as a CDF. (**C**) Change in translation efficiency (ΔTE) between *Zfp36* KO and WT was calculated as a delta between log2(KO/WT) from ribosome profiling and RNAseq datasets. The distribution of ZFP36 targets in mRNAs ranks by ΔTE is shown (left), along with normalized enrichment scores and FDRs from GSEA (right). Intron-bound mRNAs are shown as a representative gene set that show no enhanced TE in *Zfp36* KO cells. (**D**) Normalized coverage of ribosome profiling reads for *Tnf* and *Ifng* mRNAs in *Zfp36* KO and WT cells, with p-values from binomial tests. (**E**) Normalized coverage of ribosome profiling reads across all mRNAs for *Zfp36* KO and WT cells. Ribosome profiling data are a compilation of two experiments, with four total biological replicates per conditions.

DOI: https://doi.org/10.7554/eLife.33057.009

The following figure supplement is available for figure 3:

**Figure supplement 1.** Analysis of ZFP36 translational control by ribosome profiling.

DOI: https://doi.org/10.7554/eLife.33057.010

## ZFP36 negatively regulates T-cell activation kinetics

ZFP36 target mRNAs pointed to multilayered control of T cell function, including its reported regulation of effector cytokines (e.g. *Il2*, *Ifng*, *Tnf*, *Il4*, *Il10*; *Figure 4—figure supplement 1*). Novel targets spanned direct components of the TCR complex (e.g. *CD3d*, *CD3e*), co-stimulatory and co-inhibitory molecules (e.g. *Cd28*, *Icos*, *Ctla4*), TCR-proximal signaling (*Fyn*, *Sos1*, *Akt1*), and transcriptional response (e.g. *Fos*, *Nfatc1*, *Nfkb1*). As an unbiased assessment, we examined the distribution of ZFP36 targets in high-resolution gene expression time courses of CD4 + T-cell activation (*Yosef et al., 2013*). ZFP36 targets were enriched for mRNAs, like its own, that were rapidly induced after T-cell activation, and targets were depleted among mRNAs with stable expression or delayed induction (*Figure 4A*). Gene Ontology (GO) enrichments spanned many basic metabolic and gene regulatory functions, in addition to signal transduction, cellular proliferation, and apoptosis (*Figure 4B*; *Supplementary file 2*).

Functional clustering of ZFP36 targets in proliferation and apoptosis prompted us to investigate potential regulation of T cell proliferation. In thymidine incorporation assays, naïve *Zfp36* KO CD4 +T cells showed greater proliferation than WT from 16 to 24 hr post-activation (*Figure 4C*). Similar results were obtained with CD8 +T cells (*Figure 4—figure supplement 2A*). This increase reflected decreased apoptosis (*Figure 4D*) and increased numbers of proliferating cells (*Figure 4E*) in KO versus WT cultures. We examined whether an action on IL-2 might account for enhanced proliferation, as increased IL-2 production in *Zfp36* KO T cells has been reported (*Ogilvie et al., 2005*), and our HITS-CLIP data confirmed direct interaction. *Zfp36* KO T cells proliferated more than WT both in the presence of excess recombinant IL-2 or neutralizing anti-IL-2 antibody, as well as in different Th polarizing conditions, indicating the effect is not solely IL-2-dependent (*Figure 4—figure supplement 2B–C*). In summary, T cells from *Zfp36* KO mice show enhanced proliferation shortly after activation under all conditions examined.

Anti-CD3 is not a physiologic stimulation, so we next examined proliferative responses to MHC-peptide-mediated TCR binding. First, we bred WT and *Zfp36* KO mice with a transgenic, class-II restricted TCR specific for a β-galactosidase-derived antigen (BG2). BG2 transgenic *Zfp36* KO cells showed greater proliferation than WT across a broad titration of cognate peptide, but not irrelevant peptide (*Figure 4F*). Second, *Zfp36* KO T cells also showed greater proliferation than WT in response to allogeneic DCs (*Figure 4G*). Therefore, *Zfp36* KO cells show an exaggerated proliferative response upon MHC-peptide stimulation over a range of signal strengths.

Analysis of canonical T-cell activation markers revealed enhanced induction of CD69 and CD25 in *Zfp36* KO versus WT cells over the first 24 hr post-activation (*Figure 4H*). At 40 hr post-activation, a greater proportion of *Zfp36* KO versus WT had transitioned from a naïve to effector surface phenotype (*Figure 4I*). Notably, thymidine incorporation data showed enhanced proliferation of *Zfp36* KO cells early after activation, but similar rates in *Zfp36* KO and WT cells after 24 hr (*Figure 2C*). Collectively, these results show accelerated activation kinetics in the absence of ZFP36.

## ZFP36 regulation of T-cell activation is cell-intrinsic

The accelerated activation of *Zfp36* KO T cells could in principle reflect the activity of other cell subsets or inflammatory signals in *Zfp36* KO mice. To test for a T cell-intrinsic function, we generated mixed bone marrow (BM) chimeras, allowing isolation of WT and KO T cells that develop in the same in vivo milieu (*Figure 5A*). Naïve *Zfp36* KO T cells sorted from chimeras showed greater proliferation than WT 24 hr post-activation, indicating cell-intrinsic effects (*Figure 5B*). To assess the potential impact of secreted factors, chimera-derived WT and *Zfp36* KO cells were re-mixed 1:1 ex vivo. Here, differences between *Zfp36* KO and WT cells were still significant, but blunted compared to separate cultures. This result indicates cross-regulatory effects between WT and KO cells through secreted or surface factors, but these do not fully account for the observed differences. Interestingly, the reduced proliferation of *Zfp36* KO cells in mixed (*Figure 5B*, right panel) versus separate (left panel) cultures indicate that WT cells can exert a restraining effect on KO cells. Thus, accelerated activation in *Zfp36* KO cells may in part reflect compromised autoregulatory and/or suppressive functions. Three days after activation, mixed cultures remained skewed in favor of *Zfp36* KO cells, again confirming accelerated expansion (*Figure 5C*). These results show that ZFP36 regulation of T-cell activation is cell-intrinsic, and that ZFP36 normally functions to restrain T -cell activation.

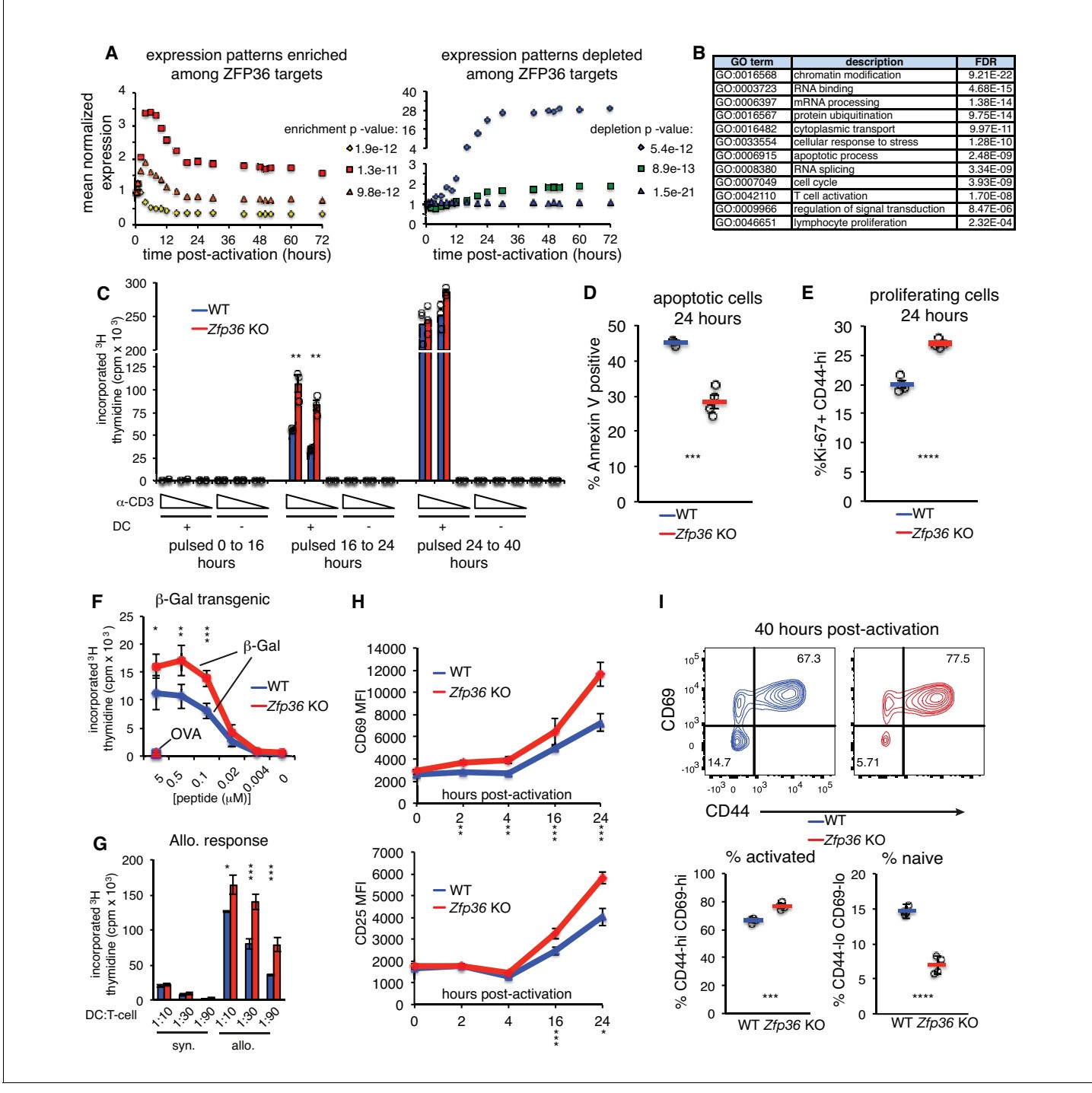

**Figure 4.** ZFP36 regulates T-cell activation kinetics. (A) Gene expression patterns from a T-cell activation time course (*Yosef et al., 2013*) were partitioned by k-means, and enrichment of ZFP36 3'UTR and CDS targets was determined across clusters (Fisher's Exact Test). Mean expression of genes in the three clusters most enriched (left) or depleted (right) for ZFP36 targets is plotted. (B) Enriched GO terms among ZFP36 HITS-CLIP targets (full results in *Supplementary File 2*). (C) Proliferation of naïve CD4 +*Zfp36* KO and WT T cells in the indicated time windows after activation, measured by [3]H-thymidine incorporation (D) Fractions of apoptotic annexin-V+ and (E) proliferating Ki67 +CD4+T cells 24 hr post-activation. Mean ± S.E.M. is shown; circles are individual mice (n = 3–4 per genotype). (F) Proliferation of BG2 TCR-transgenic CD4 +T cells cultured with DCs pulsed with cognate (β-gal) or irrelevant (OVA) peptide. Mean ± S.E.M. is shown (n = 5 mice per genotype). (G) Proliferation of CD4 +T cells co-cultured with syngeneic (C57BL6/J) or allogeneic (Balb-c) DCs. Mean ± S.E.M. of three replicate cultures is shown. (H) Levels of CD69 and CD25 after activation of *Zfp36* KO and WT naïve CD4 +T cells. Mean ± S.E.M. is shown (n = 3–4 mice per genotype). (I) Naïve and effector subsets 40 hr post-activation in *Zfp36* KO and WT

*Figure 4 continued on next page*

*Figure 4 continued*

CD4 +T cells. Representative plots are shown (top), along with mean ± S.E.M and circles for individual mice (n = 4 per genotype). For (**C–I**), results of two-tailed t-tests: *=p < 0.05; **=p < 0.01; ***=p < 0.001. Data are representative of three (**H**) or two (**C–G, I**) independent experiments.

DOI: https://doi.org/10.7554/eLife.33057.011

The following figure supplements are available for figure 4:

**Figure supplement 1.** ZFP36 targets regulate T-cell activation.

DOI: https://doi.org/10.7554/eLife.33057.012

**Figure supplement 2.** ZFP36 regulates early activation across T cell lineages.

DOI: https://doi.org/10.7554/eLife.33057.013

## Downstream effects of ZFP36 regulation

The efficient in vitro responses of *Zfp36* KO T cells suggest they are functional, but respond with altered kinetics. To examine the downstream consequences of this differential regulation, HITS-CLIP and RNAseq analyses were done in Th1-skewed CD4 +T cells 3 days after activation. ZFP36 binding site features in cells activated for 3 days were similar to ones identified at 4 hr (*Figure 6—figure supplement 1A–B*; *supplementary file 3*), but results from transcriptome profiling were strikingly different at these two time points (*Figure 6A*). First, in contrast to subtle effects observed at the 4 hr time point, many transcripts showed highly divergent expression in *Zfp36* KO versus WT T cells 3

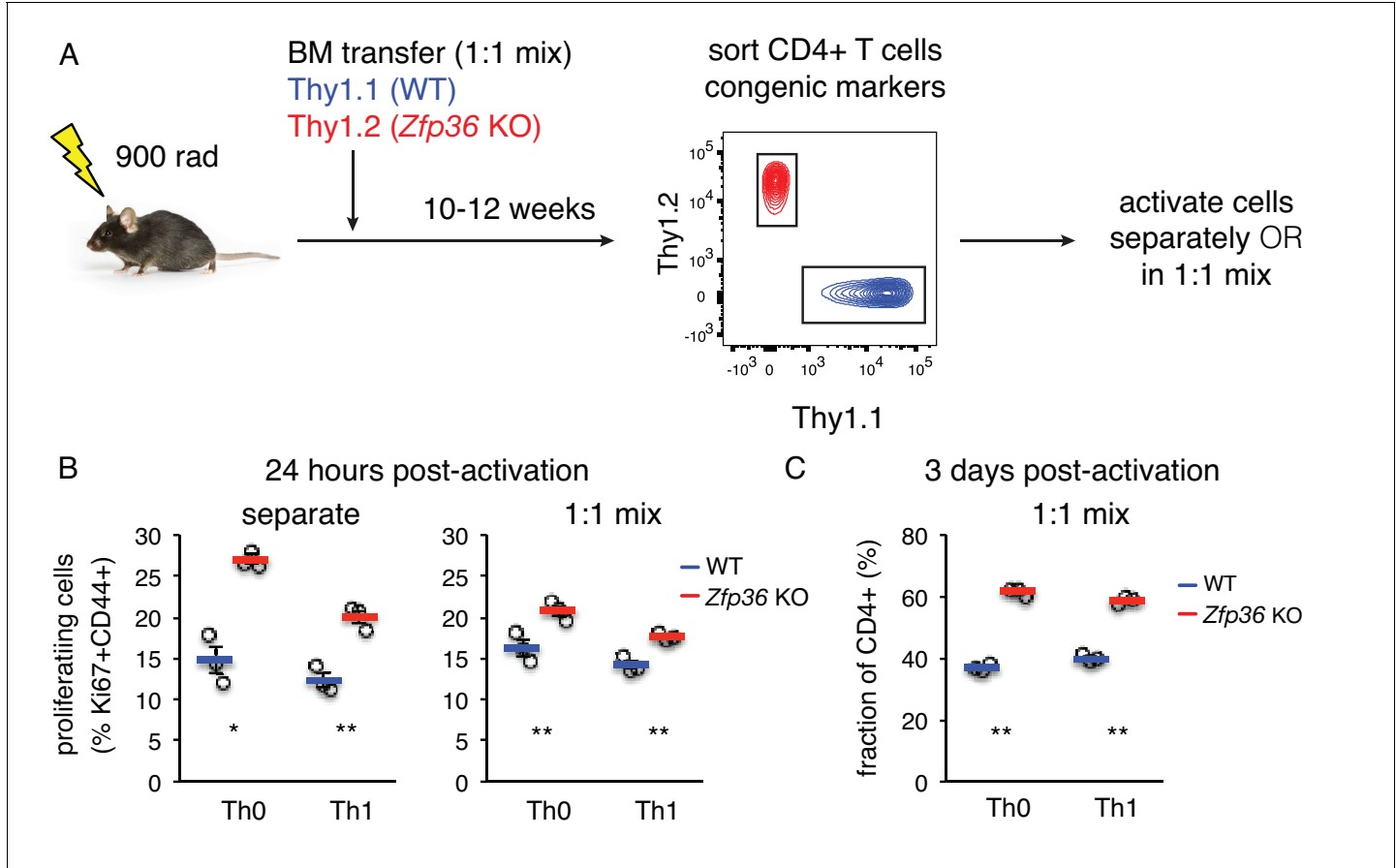

**Figure 5.** ZFP36 regulation of T cell activation kinetics cell-intrinsic. (**A**) Lethally irradiated mice were reconstituted with congenically marked WT and *Zfp36* KO BM to generate mixed chimeras. 10–12 weeks after reconstitution, naïve CD4 +WT and *Zfp36* KO T cells were sorted, then activated ex vivo separately or mixed 1:1. (**B**) Proliferating Ki67 +cells were measured 24 hr after activating naïve CD4 +T cells under Th0 or Th1 conditions. (**C**) Cultures with a 1:1 starting ratio of naïve WT and *Zfp36* KO CD4 +T cells were examined 3 days post-activation. Data from one experiment of two performed are shown.

DOI: https://doi.org/10.7554/eLife.33057.014

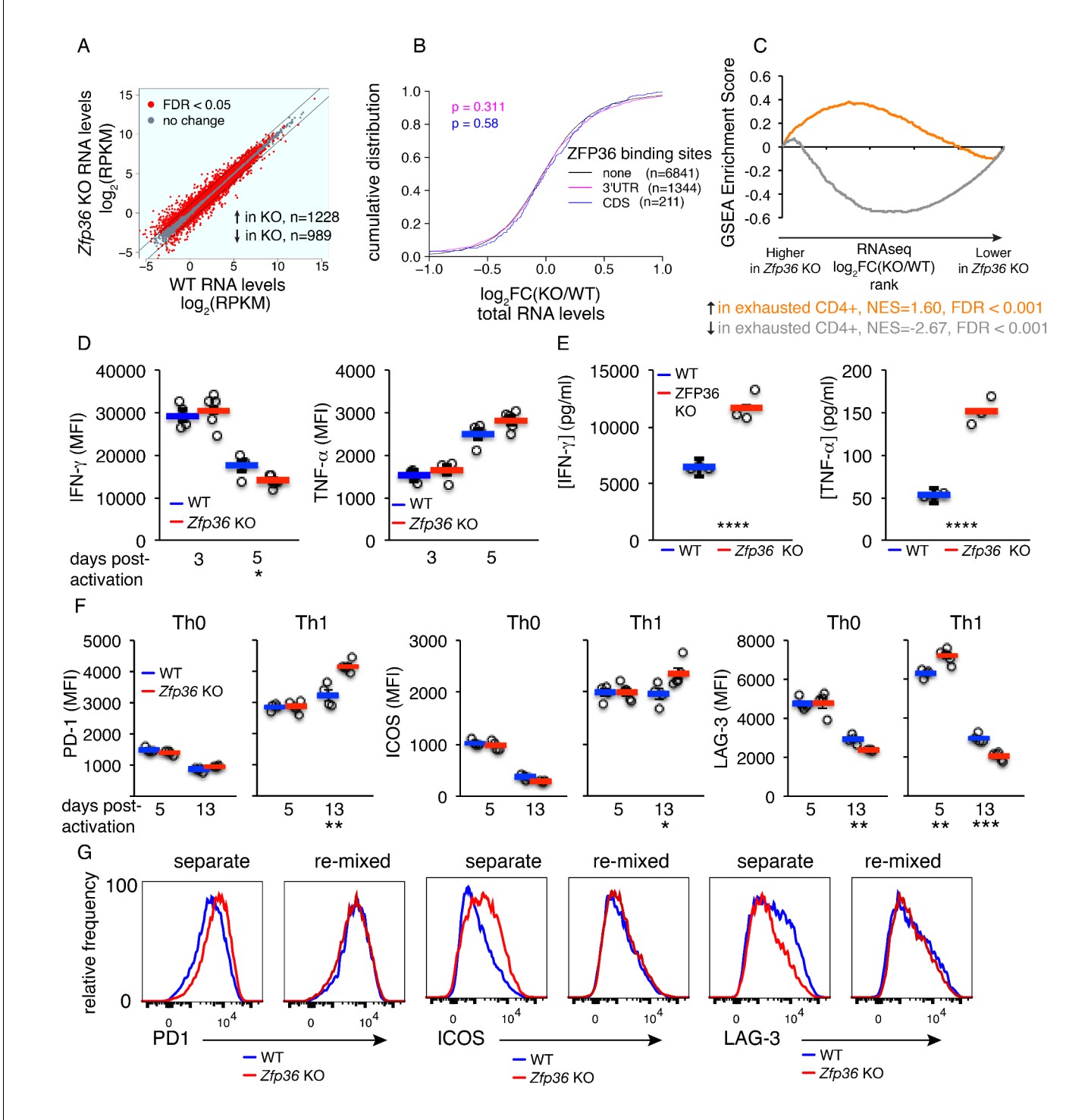

**Figure 6.** Accelerated signs of in vitro T cell exhaustion in absence of ZFP36. (**A**) Log2-transformed RPKM values from *Zfp36* KO versus WT CD4 +Th1 cell RNAseq 72 hr post-activation, with red indicating differential expression (FDR < 0.05). Lines mark 2-fold changes. RNAseq data represent one experiment with three biological replicates per condition. (**B**) Log2-transformed fold-changes (KO/WT) plotted as a CDF, for mRNAs with 3'UTR, CDS, or no significant ZFP36 HITS-CLIP. Numbers of mRNAs in each category (**n**) and p-values from KS tests are indicated. (**C**) The gene expression profile in *Zfp36* KO CD4 +T cells 72 hr post-activation was compared to reported profiles of CD4 +T cell exhaustion using GSEA. Upregulated (orange) and downregulated (gray) gene sets in exhausted T cells showed strong overlap with corresponding sets from *Zfp36* KO T cells (FDR < 0.001, hypergeometric test). (**D**) IFN-γ and TNF-α measured by ICS 3 and 5 days after activation of naïve CD4 +T cells. (**E**) IFN-γ and TNF-α in culture supernatants 3 and 5 days after activation of naïve CD4 +T cells. (**F**) PD-1, ICOS, and LAG-3 expression 5 and 13 days after activation under Th0 or Th1

*Figure 6 continued on next page*

*Figure 6 continued*

conditions. (D–F) show mean ± S.E.M.; circles are individual mice (n = 3–5 per genotype). (G) Measurements as in (F) for *Zfp36* KO and WT CD4 +T cells derived from mixed BM chimeras. Cells were activated under Th1 conditions for 13 days, either separately or mixed 1:1. For (D–G), one representative experiment of two performed is shown. Results of two-tailed t-tests: *=p < 0.05; **=p < 0.01; ***=p < 0.001; ****=p < 0.0001.

DOI: https://doi.org/10.7554/eLife.33057.015

The following figure supplements are available for figure 6:

**Figure supplement 1.** Analysis of ZFP36 function 3 days after T cell activation.

DOI: https://doi.org/10.7554/eLife.33057.016

**Figure supplement 2.** Dysfunction of *Zfp36* KO T cells at late time points.

DOI: https://doi.org/10.7554/eLife.33057.017

days after activation (*Figure 6A*). However, these differences were not correlated to ZFP36 HITS-CLIP binding at 72 hr (*Figure 6B*). Thus, the absence of ZFP36 in the early phases of T -cell activation can lead to significant secondary effects downstream.

GSEA with RNAseq data from Th1 *Zfp36* KO cells 3 days post-activation showed reduced activity of transcription factors driving proliferation (e.g. Myc and E2F; *Figure 6—figure supplement 2A*) and strong overlap with previously described signatures of T cell exhaustion (*Figure 6C*; [*Crawford et al., 2014*]). Consistent with a late activated and exhausted phenotype in *Zfp36* KO cells, the enhanced production of IFN-γ at early time points (*Figures 2A* and *3D*) gave way to comparable production by day three and reduced production by day 5 (*Figure 6D*). Moreover, TNF-α production was not significantly different 3 or 5 days post-activation (*Figure 6D*), despite large differences early on (*Figures 2A* and *3D*). In summary, relieved translational control drives elevated cytokine production in *Zfp36* KO cells early post-activation, but enhanced production dissipates downstream due to more rapid expansion and exhaustion. The net result, reflecting both dysregulated cytokine production and more rapid culture expansion (*Figures 4* and *5*), is a higher accumulation of IFN-γ and TNF-αin *Zfp36* KO culture supernatants 72 hr post-activation (*Figure 6E*).

We examined co-inhibitory and co-stimulatory checkpoint proteins that are linked to T cell exhaustion, and found elevated expression of PD-1 and ICOS at late time points in *Zfp36* KO cells, and more rapid peaking of LAG-3 (*Figure 6F*). Interestingly, these effects were observed in Th1 but not Th0 conditions, suggesting a dependence on Th1 cytokines. To test this dependence, and to examine whether elevated receptor expression was T cell-intrinsic, we analyzed cells derived from mixed BM chimeras. This analysis confirmed differential, T cell-intrinsic expression of these receptors (*Figure 6G*). However, re-mixing WT and KO cells ex vivo neutralized these differences, indicating they are driven by secreted factors. We tested whether recombinant IFN-γ, supplemented at levels measured in KO cultures, could cause elevated receptor expression in WT Th1 cells, and found it promoted ICOS but not PD-1 upregulation (*Figure 6—figure supplement 2B*). These results indicate that Th1 cytokines, including but not only IFN-γ, can drive an exhaustion-like phenotype. The absence of ZFP36 promoted this phenotype in vitro, due to more rapid activation and expansion coupled with greater accumulation of Th1 cytokines.

## ZFP36 regulates antiviral immunity

The accelerated response of *Zfp36* KO T cells, and the potential for accelerated exhaustion, led us to examine the effects of ZFP36 regulation in vivo. We first determined that naïve *Zfp36* KO mice had normal T cell levels in peripheral blood and no defects in thymocyte development (*Figure 7—figure supplement 1A–B*). Total splenocytes, including T cells, were slightly reduced in *Zfp36* KO versus WT mice (*Figure 7—figure supplement 1C*), but proportions of total CD4 + and CD8+T cells were normal (*Figure 7—figure supplement 1D*). Proportions of naïve CD4 +T cells were also normal in KO mice, and naïve CD8 +T cells only slightly decreased. (*Figure 7—figure supplement 1E*). Levels of CD25-hi CD4 +cells were not significantly different in spleens of WT and KO mice, consistent with similar levels of natural Tregs (*Figure 7—figure supplement 1F*). FoxP3 expression was not examined directly ex vivo, but in vitro induction of Tregs from naïve cells CD4 +T cells, enumerated in FoxP3-GFP mice, was not different between WT and KO (*Figure 7—figure supplement 1G*; [*Haribhai et al., 2007*]).

To examine the effector T cell populations present in spleen, splenocytes were stimulated directly ex vivo with phorbol myristate acetate (PMA) and ionomycin. More CD4 + and CD8+T cells

produced IFN-γin KO versus WT splenocytes, but levels of IL-4 and IL17A production were comparable (*Figure 7—figure supplement 1H*). Therefore, greater numbers of Th1 cells are present at steady state in *Zfp36* KO mice. To examine whether loss of ZFP36 causes an intrinsic disposition to the Th1 fate, skewing of sorted, naïve CD4 +T cells was examined. These analyses showed indistinguishable differentiation of Th1 and Th17 subsets ex vivo in WT and *Zfp36* KO cells (*Figure 7—figure supplement 1I*). The greater accumulation of Th1 cells in vivo may therefore reflect a response to factors not replicated in vitro, or the effects of additional cell types.

The lymphocytic choriomeningitis virus (LCMV) Armstrong strain causes an acute infection leading to massive T cell expansion and viral clearance in 8–10 days (*Dutko and Oldstone, 1983*). Using MHC-tetramers, we observed accelerated expansion and recession of virus-specific CD4+ (*Figure 7A*) and CD8+ (*Figure 7B*) T cells in *Zfp36* KO versus WT mice in peripheral blood. This result was confirmed in independent experiments focused on early time points post-infection (p.i.), where virus-specific T cells in *Zfp36* KO mice showed earlier expansion and more rapid upregulation of CD69 (*Figure 7C–D*). Enumeration of virus-specific T cells in spleen mirrored dynamics in blood; levels were greater in *Zfp36* KO animals 6 days p.i., but marginally lower by day 10, consistent with more rapid expansion and resolution (*Figure 7E–F*). Levels of memory T cells day 40 p.i. were similar in *Zfp36* KO and WT mice. In summary, antigen-specific T cell response is clinically functional but accelerated in *Zfp36* KO mice during viral infection.

Stimulation with LCMV peptides ex vivo revealed higher rates of IFN-γ and TNF-αproduction in *Zfp36* KO versus WT CD4+ (*Figure 7G*) and CD8 +T cells 6 days p.i. (*Figure 7—figure supplement 2A*). Numbers of cytokine-producing cells were proportional to LCMV-specific tetramer +cells (*Figure 7E–F*). However, levels of IFN-γand TNF-α protein were significantly greater in CD4 +*Zfp36* KO cytokine-producing cells versus WT (*Figure 7H*), and TNF-α levels were also higher for CD8 +cells (*Figure 7—figure supplement 2B*). In addition, 'bifunctional' IFN-γ+TNF-α+T cells were more frequent in *Zfp36* KO mice, even when normalized to frequencies of tetramer +cells (*Figure 7I* and *Figure 7—figure supplement 3C*).

Strikingly, LCMV genomic RNA in spleen was ~10 fold lower day 6 p.i. in *Zfp36* KO versus WT animals, consistent with more rapid clearance of LCMV infection (*Figure 7J*). Viral load correlated inversely with levels of tetramer +CD4+ and CD8+T cells in both *Zfp36* KO and WT mice, consistent with the established role of T cell response in LCMV clearance (*Figure 7—figure supplement 2D*). To examine whether the accelerated LCMV-specific T cell response in *Zfp36* KO mice may be T cell-intrinsic, infections were repeated in mixed BM chimeras. Irradiated recipient mice were re-constituted with a 1:1 mix of congenically marked WT or *Zfp36* KO BM cells (*Figure 7—figure supplement 3A*). Ten weeks after re-constitution, pre-infection baseline measurements showed a significantly greater expansion of *Zfp36* KO T cells versus WT in blood (*Figure 7—figure supplement 3B*). However, the kinetics of WT and KO T cell response in these animals upon LCMV infection were indistinguishable (*Figure 7—figure supplement 3C–D*). Therefore, in a mixed environment in vivo, *Zfp36* KO and WT T cells show similar kinetics upon viral challenge. Notably, maximum T cell expansion was observed in mixed chimeras 7–8 days p.i., which was intermediate to the maxima observed in *Zfp36* KO (6 days) and WT (10 days) mice.

Collectively, these data demonstrate a remarkable enhancement of anti-viral immunity in the setting of reduced ZFP36 family activity in vivo. This enhancement was marked by an accelerated T cell response and enhanced production of effector cytokines and, based on mixed chimera experiments, may involve immune cell types in addition to T cells.

## Discussion

Immune response requires rapid, adaptable gene regulation—features uniquely suited to post-transcriptional control. Our studies illuminate a role for ZFP36 RNA binding proteins in controlling the pace of T cell response, a crucial dimension of adaptive immunity, and tie this to the effectiveness of anti-viral responses in vivo.

ZFP36 and ZFP36L1 expression are rapidly induced upon T-cell activation, and gradually recede thereafter. While transcriptional induction has been previously established, our detection of *Zfp36* and *Zfp36l1* mRNAs in naïve T cells, in the absence of detectable protein, indicates post-transcriptional regulation of their mRNAs in T cells. Moreover, paralog auto- and cross-regulation are likely, as *Zfp36* and *Zfp36l1* mRNA 3'UTRs possess robust HITS-CLIP binding sites. We did not detect

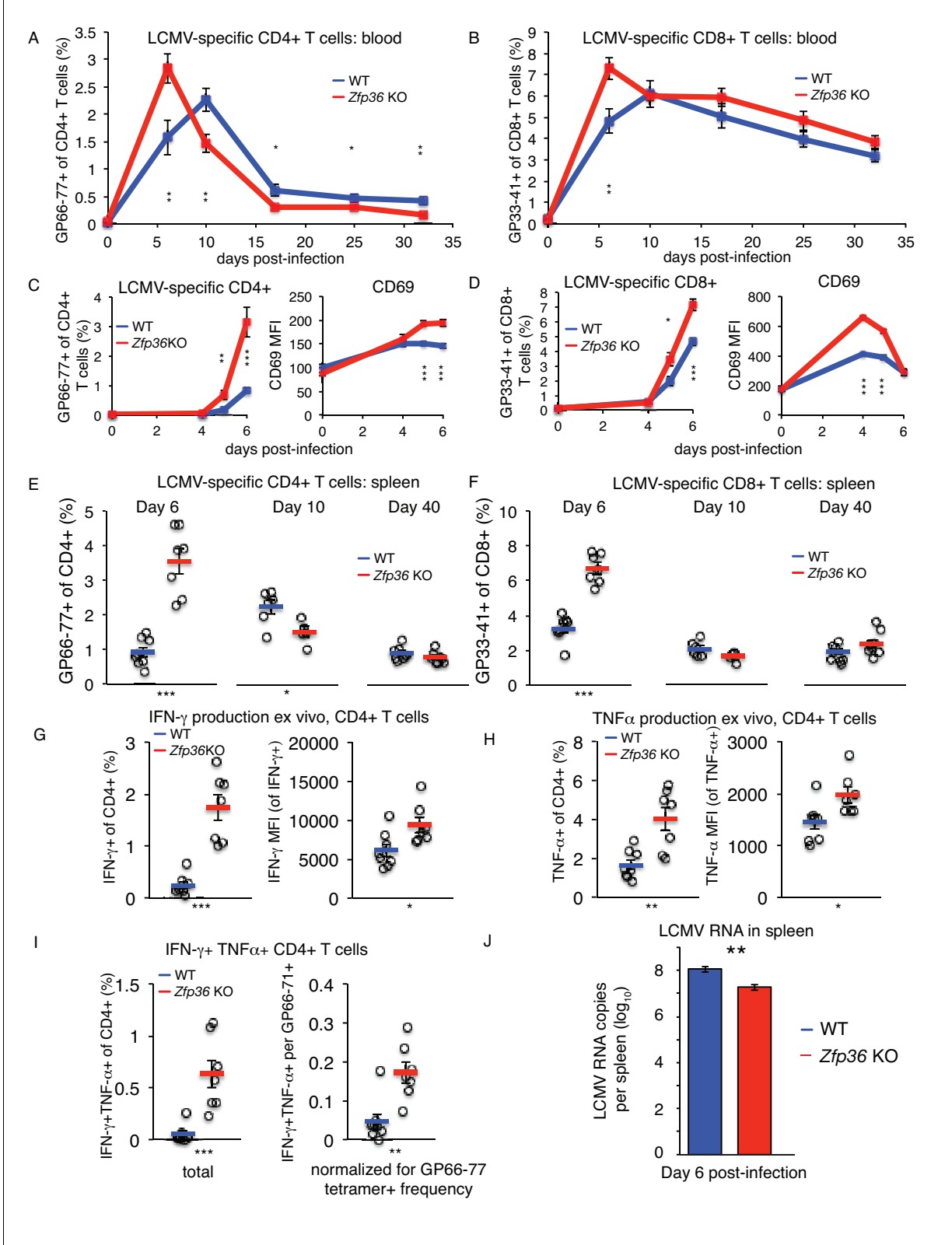

**Figure 7.** ZFP36 regulates anti-viral immunity. (**A**) Virus-specific CD4 + or (**B**) CD8 +T cells were tracked in peripheral blood using MHC-tetramers after LCMV Armstrong infection (n = 8–9 mice per genotype). (**C**) Virus-specific CD4 +T cells and CD69 expression on CD4 +T cells in peripheral blood at early time points post-infection (p.i.) (n = 7–8 mice per genotype). (**D**) Virus-specific CD8 +T cells and CD69 expression on CD8 +T cells in peripheral blood at early time points p.i. (n = 7–8 mice per genotype). (**E**) Virus-specific CD4 +and (**F**) CD8 +T cells in spleen after LCMV infection (n = 5–8 mice

*Figure 7 continued on next page*

*Figure 7 continued*

per genotype). (G) Fraction of CD4 +T cells producing IFN-γ and TNF-α in splenic CD4 +T cells 6 days p.i., after ex vivo stimulation with GP66-77 peptide (n = 7–8 mice per genotype). (H) Levels of IFN-γ and TNF-α(gated on cytokine-producing CD4 +cells) 6 days p.i. after ex vivo stimulation with GP66-77 (n = 7–8 mice per genotype). (I) Raw percentage of bifunctional IFN-γ+TNF-α+CD4+cells in spleen 6 days p.i. after ex vivo stimulation with GP66-77 (left), or normalized to percentage of GP66-77 tetramer +cells (n = 7–8 mice per genotype). (J) Levels of LCMV genomic RNA in spleen measured by RT-qPCR (n = 9–14 per group). For (A–J), mean values ± S.E.M. are shown, with circles as individual mice. Results of two-tailed t-tests: *=p < 0.05; **=p < 0.01; ***=p < 0.001; ****=p < 0.0001. In each panel, one representative experiment of two is shown.

DOI: https://doi.org/10.7554/eLife.33057.018

The following figure supplements are available for figure 7:

**Figure supplement 1.** The T cell compartment in naïve *Zfp36* KO mice is largely normal.

DOI: https://doi.org/10.7554/eLife.33057.019

**Figure supplement 2.** ZFP36 regulates anti-viral immunity.

DOI: https://doi.org/10.7554/eLife.33057.020

**Figure supplement 3.** LCMV infection in mixed bone marrow chimeras.

DOI: https://doi.org/10.7554/eLife.33057.021

ZFP36L2 in any of our analyses, indicating it is absent or negligibly expressed under conditions examined here. However, the presence of its mRNA further suggests post-transcriptional control of ZFP36 paralog expression, and is consistent with functions in other contexts or stages of T cell function ([*Vogel et al., 2016*; *Hodson et al., 2010*]).

Definitive determination of ZFP36 targets in T cells by HITS-CLIP, coupled with transcriptome and ribosome profiling studies, revealed that ZFP36 attenuates T-cell activation by suppressing the abundance and translation of its mRNA targets. The correlation of ZFP36 binding with reduced mRNA abundance is consistent with reports that ZFP36 can destabilize target mRNAs by recruiting degradation factors (*Fabian et al., 2013*; *Lykke-Andersen and Wagner, 2005*). However, our ability to stratify the relative magnitude of ZFP36 binding using CLIP resolved a more complex trend, with highly robust 3'UTR binding sites (top 20%) showing no detectable correlation with RNA abundance (*Figure 1—figure supplement 3C*). This non-uniform trend was observed for 3'UTR but not CDS targets, and affected mRNA abundance but not ribosome association. Importantly, effects on protein levels in the absence of changes in mRNA abundance were confirmed independently for robust 3'UTR binding targets *Tnf*, *Ifng*, and *Cd69*. It is possible that different degrees of ZFP36 association in vivo elicit distinct functional outcomes, through differential RNP localization or downstream effector recruitment. Notably, ZPF36 CLIP showed a broad MW range of ZFP36-RNA complexes with distinct biochemical properties including stability to heat, detergent, and high salt. While the current studies did not uncover distinct mRNA targets across this range, more detailed biochemical studies will be necessary to clarify potentially distinct ZFP36 complexes in vivo, and their potentially distinct roles in different cell types of the immune system.

We further present evidence that ZFP36 suppresses translation of its target mRNAs in T cells. Endogenous targets and exogenous reporters showed greater ZFP36-dependent suppression of protein versus RNA levels (*Figure 2*), and ribosome profiling in primary T cells confirmed direct effects on translation (*Tao and Gao, 2015*; *Qi et al., 2012*). The strongest effects were linked to a novel class of AREs in coding sequence, uncovered with ZFP36 binding maps. CDS sites correlated with repressed RNA abundance and translation, but a greater level of repression was evident in ribosome association (*Figure 3B–D*). Intriguingly, some ZFP36 CLIP reads in CDS sites spanned exon-intron boundaries, indicating these associations can form prior to pre-mRNA splicing in the nucleus (*Figure 1—figure supplement 2E*). The identification of AREs in the CDS and the possibility of resulting translation control pre-programmed in the nucleus point to novel, unexplored regulatory strategies. Notably, these results differ significantly from iCLIP analyses in macrophages using exogenous GFP-tagged ZFP36, where only 3'-UTR sites correlated with target repression, and only for a ZFP36 construct with mutated MK2 phosphorylation sites (*Tiedje et al., 2016*). In those studies, the WT ZFP36 construct showed negligible repressive effects, contrasting with our data in 293 and T cells, and data from other contexts (*Tao and Gao, 2015*; *Ogilvie et al., 2009*). However, iCLIP data for the transduced WT ZFP36 showed low 3'UTR binding (23%), high intergenic binding (38%), and a preference for GU-rich motifs, diverging sharply from our analysis of endogenous ZFP36 and prior in vivo and in vitro characterizations (*Brewer et al., 2004*; *Worthington et al., 2002*). These

differences may reflect distinct ZFP36 phosphorylation, and hence regulatory outcomes, or as yet undefined variables related to the different cellular context. A direct comparison is further confounded by the use of exogenous, transduced constructs in macrophage experiments, in contrast to our analysis of endogenous proteins in T cells. Importantly, the methods and reagents developed here allow for systematic analysis of endogenous ZFP36 proteins in future global and cell-type-specific investigations addressing these issues.

The similarity of RNA-binding maps covering both ZFP36 and ZFP36L1 (WT cells) or ZFP36L1 alone (*Zfp36* KO cells) supports redundancy of ZFP36 paralogs, a likely source of robustness in immune regulation. *Zfp36* KO cells are thus likely a partial loss-of-function due to robust ZFP36L1 expression, a notion consistent with the relatively subtle regulatory effects on RNA abundance and translation. Phenotypically, loss of *Zfp36* led to accelerated activation of mature T cells, but not uncontrolled proliferation or impaired development, which may again may reflect a partial loss of pan ZFP36 activity. Indeed, a prior study reported no effects when *Zfp36l1* was deleted in T cells, but drastic dysregulation of thymocyte proliferation upon loss of both *Zfp36l1 and Zfp36l2* (*Hodson et al., 2010*). These studies indicate total paralog dosage is critical, but also suggest the importance of the specific balance of ZFP36 paralogs in a defined context. Improved cell profiling methods, such as cell-type-specific tagging of RBPs, may illuminate these complexities in future studies.

The mRNA targets defined by ZFP36 HITS-CLIP span from surface molecules engaged in the earliest steps in T-cell activation to downstream signaling and transcriptional effectors. Targets were strongly enriched for regulation of proliferation and apoptosis, extending prior reports that the ZPF36 family regulates proliferation in early T- and B-cell development and cancer. In each case, the reported mechanism was distinct, spanning regulation of Notch, G1/S phase transition, and Myc, respectively (*Galloway et al., 2016*; *Hodson et al., 2010*; *Rounbehler et al., 2012*). ZFP36 HITS-CLIP identified all of these target pathways in T cells, consistent with a central function in controlling cell proliferation. However, the phenotype of *Zfp36* KO T cells is novel and distinct, leading not to uncontrolled proliferation, but to accelerated effector response and resolution. The global effects of ZFP36 repression on RNA abundance and translation were widespread but subtle, and spanned many layers of T cell function. Functional validation of novel ZFP36 targets, including T-cell activation marker *Cd69* and apoptosis regulator *Bcl2*, suggest factors that likely contribute to this regulation. Analyses with cells sorted from mixed BM chimeras showed that the enhanced activation in *Zfp36* KO cells is T cell-intrinsic (*Figure 5*). Experiments in which these sorted cells were re-mixed ex vivo exhibited blunted differences, as compared to cells cultured separately (*Figure 5B*). These results indicate a role for both intracellular and secreted factors in exerting ZFP36 regulatory effects. More broadly, the combined picture of our genomic and functional data are one of many functionally diverse targets contributing to a multifaceted, finely tuned response, which is a hallmark of post-transcriptional regulatory control.

Analyses at later time points after activation in Th1 conditions revealed widespread dysregulation of *Zfp36* KO cells, with a gene expression signature and surface phenotype resembling T cell exhaustion (*Figure 6*; [*Crawford et al., 2014*]). Notably, and in contrast to analyses early post-activation, changes in RNA abundance 3 days post-activation were not correlated to direct ZFP36 binding. Therefore, secondary effects of ZFP36 loss pre-dominate as T cell expansion and differentiation progress in these settings. Notably, altered expression of co-inhibitory and –stimulatory receptors in *Zfp36* KO cells was specific to Th1 cells (*Figure 6F*), and mixing experiments confirmed dependence on secreted factors (*Figure 6G*; *Figure 6—figure supplement 2B*). Thus, a dysregulated Th1 secretion profile including but not limited to elevated IFN-γ can push T cells to a state resembling exhaustion in vitro. More broadly, these results highlight the crucial point that subtle, but direct, regulation of a broad range of targets early after T-cell activation by ZFP36 can have striking downstream consequences in a rapidly expanding cell population.

In vivo studies of acute viral infection showed accelerated expansion and recession of virus-specific CD4 + and CD8+T cells in *Zfp36* KO animals. *Zfp36* KO T cells had higher levels TNF-α and IFN-γ protein expression than WT after peptide stimulation, and more 'bi-functional' TNF-α/IFN-γ co-producing cells, thought to be important for anti-viral immunity (*Crawford et al., 2014*). More rapid T cell expansion coincided with lower accumulated viral titers (or more rapid clearance) in *Zfp36* KO animals, indicating ZFP36 regulates anti-viral immunity. In mixed BM chimeras, the response kinetics of WT and *Zfp36* KO T cells to LCMV infection were indistinguishable. This result

raises the possibility that other cell types and pathways, such as antigen presenting cells, antibody response, or innate immune regulators, could contribute to the accelerated anti-viral T cell response in *Zfp36* KO mice. It is noteworthy that the point of maximum T cell expansion in these chimeras was intermediate to that of *Zfp36* KO and WT mice. This result resembles our previous observation that WT and *Zfp36* KO cells sorted from BM chimeras show blunted differences in activation kinetics when re-mixed ex vivo, as compared to cells cultured separately (*Figure 5B*). Collectively, these data suggest that *Zfp36* KO and WT cells exhibit cross-regulatory effects in a mixed environment that may obscure or complement cell-intrinsic differences. Given that many prominent ZFP36 targets encode secreted factors, such cross-regulatory effects are a virtual certainty.

Collectively, these in vivo studies demonstrate an accelerated T cell response to viral infection in the absence of ZFP36, which may reflect the heightened activity of multiple cell types in addition to T cells. Regardless of the initiating mechanism, the observation of enhanced LCMV clearance is likely T cell-dependent, given the central role of T cells in LCMV immunity (*Matloubian et al., 1994*). Accordingly, we observed a quantitative, negative correlation between viral load and antigen-specific T cell levels in vivo (*Figure 7—figure supplement 2D*). Importantly, the enhanced anti-viral response in the chronic absence of ZFP36 in KO mice is accompanied by spontaneous inflammation and autoimmunity that worsen with age. Our results starkly illustrate the delicate balance of protective immunity against destructive inflammation, and reveal post-transcriptional regulation by RBPs as central to this trade-off.

Starting from transcriptome-wide RNA binding maps in T cells, we uncovered a crucial function for ZFP36 proteins in regulating adaptive immunity. These data suggest carefully titrated inhibition of ZFP36 might serve as a pharmacologic strategy in contexts where accelerated T cell response to challenge is desirable. Our in vivo LCMV studies demonstrate acute viral infection as one context, but application to other intracellular pathogens warrants investigation. Moreover, the ability to activate T cells to target tumor antigens and the clinical utility of checkpoint inhibitors raise the possibility of exploring ZFP36 inhibition to enhance tumor immunity. ZPF36 HITS-CLIP identified many targets central to these strategies, including *Cd274* (PD-L1), *Pdcd1l2* (PD-L2), *Icos*, *Cd27*, *Cd28*, *Ctla-4*, *Btla*, and *Lag3*, suggesting a means for concerted regulation. The autoimmune phenotype of the *Zfp36* KO mouse highlights an important caveat, common to parallel issues seen with clinical use of checkpoint inhibitors. The tools for cell-type-specific analysis of ZFP36, its targets, and its inhibition now exist to investigate and refine this balance.

## Materials and methods

**Key resources table**

| Reagent type (species) or resource | Designation | Source or reference | Identifiers | Additional information |
|---|---|---|---|---|
| Gene (*Mus musculus*) | *Zfp36* | NA | Entrez ID: 22695 | |
| Gene (*M. musculus*) | *Zfp36l1* | NA | Entrez ID: 12192 | |
| Gene (*M. musculus*) | *Zfp36l2* | NA | Entrez ID: 12193 | |
| Strain (*M. musculus*), strain background (*C57BL6/J*) | C57BL6/J | Jackson Laboratory | Stock No: 000664 | |
| Strain (*M. musculus*), strain background (*C57BL6/J*) | ZFP36 KO | PMID:8630730 | | gift from P. Blackshear |
| Strain (*M. musculus*), strain background (*C57BL6/J*) | BG2 | PMID:19478869 | | gift from N. Restifo |
| Strain (*M. musculus*), strain background (*C57BL6/J*) | Thy1.1 | Jackson Laboratory | Stock No: 000406 | |
| Strain (*M. musculus*), strain background (*C57BL6/J*) | CD45.1 | Jackson Laboratory | Stock No: 002014 | |

*Continued on next page*

*Continued*

| Reagent type (species) or resource | Designation | Source or reference | Identifiers | Additional information |
|---|---|---|---|---|
| Strain (*M. musculus*), strain background (*C57BL6/J*) | FoxP3-EGFP | Jackson Laboratory | Stock No: 006769 | |
| Strain (*Lymphocytic Choriomeningitis Virus, LCMV*), strain background (*Armstrong*) | LCMV Arm | PMID:6875516 | | |
| Cell line (*Homo sapien*) | 293 T-rex | Life Technologies | Cat# R71007 | |
| Cell line (*H. sapien*) | 293T | ATCC | ATCC Cat# CRL-3216, RRID:CVCL_0063 | |
| Cell line (*M. musculus*) | J558L/GM-CSF | PMID:1460426 | | |
| Transfected construct (*M. musculus*) | pOZ-N-FH-ZFP36 | This study | | mouse *Zfp36* ORF in pOZ-N vector |
| Transfected construct (*M. musculus*) | pOZ-N-FH-ZFP36L1 | This study | | mouse *Zfp36l1* ORF in pOZ-N vector |
| Transfected construct (*M. musculus*) | pOZ-N-FH-ZFP36L2 | This study | | mouse *Zfp36l2* ORF in pOZ-N vector |
| Transfected construct | pOZ-N-FH vector | PMID:14712665 | | |
| Transfected construct | pcDNA3.1(+) | Life Technologies | Cat# V79020 | |
| Transfected construct | pcDNA3.1(+)-Acgfp1-IFNG-WT-UTR | This paper | | Acgfp1 with mouse *Ifng* 3'UTR |
| Transfected construct | pcDNA3.1(+)-Acgfp1-IFNG-WT-UTR | This paper | | Acgfp1 with mouse *Ifng* 3'UTR with *Zfp36* binding site deleted |
| Transfected construct | pcDNA5/FRT/TO | Life Technologies | Cat# V652020 | |
| Transfected construct | pcDNA5/FRT/TO/Acgfp1-TNF-WT-UTR | This paper | | Acgfp1 with mouse *Tnf* 3'UTR |
| Transfected construct | pcDNA5/FRT/TO/Acgfp1-TNF-Δ-UTR | This paper | | Acgfp1 with mouse *Tnf* 3'UTR with *Zfp36* binding site deleted |
| Transfected construct | pcDNA5/FRT/TO/Acgfp1-CD69-WT-UTR | This paper | | Acgfp1 with mouse *Cd69* 3'UTR |
| Transfected construct | pcDNA5/FRT/TO/Acgfp1-CD69-Δ-UTR | This paper | | Acgfp1 with mouse *Cd69* 3'UTR with *Zfp36* binding site deleted |
| Antibody | Rabbit anti-pan-ZFP36 RF2046 | This paper | Covance custom service | 1:2000 for Western |
| Antibody | Rabbit anti-pan-ZFP36 RF2047 | This paper | Covance custom service | 1:2000 for Western |
| Antibody | anti-Br-dU | Millipore | Millipore Cat# MAB3222; RRID:AB_11212494 | 5 µg per IP |
| Antibody | rabbit anti-TTP/ZFP36 | Sigma | Sigma-Aldrich Cat# T5327; RRID:AB_1841222 | 1:500 |
| Antibody | rabbit anti-ZFP36L1/2 (BRF1/2) | CST | Cell Signaling Technology Cat# 2119S; RRID:AB_10695874 | 1:500 |
| Antibody | mouse anti-FLAG | Sigma | Sigma-Aldrich Cat# F3165; RRID:AB_259529 | 1:500 |
| Antibody | mouse anti-FUS | Santa Cruz | Santa Cruz Biotechnology Cat# sc-47711; RRID:AB_2105208 | 1:1000 |
| Antibody | rabbit anti-FUS | Novus | Novus Cat# NB100-562; RRID:AB_10002858 | 1:10000 |

*Continued on next page*

Continued

| Reagent type (species) or resource | Designation | Source or reference | Identifiers | Additional information |
|---|---|---|---|---|
| antibody | goat anti-rabbit-IgG-680RD | LICOR | LI-COR Biosciences Cat# 925–68071; RRID:AB_2721181 | 1:25000 |
| Antibody | goat anti-rabbit-IgG-800CW | LICOR | LI-COR Biosciences Cat# 925–32211; RRID:AB_2651127 | 1:25000 |
| Antibody | goat anti-mouse-IgG-800CW | LICOR | LI-COR Biosciences Cat# 925–32210; RRID:AB_2687825 | 1:25000 |
| Antibody | anti-CD4-PerCP-Cy5.5 | BD Biosciences | BD Biosciences Cat# 550954; RRID:AB_393977 | 1:400 |
| Antibody | anti-CD25-PE/Cy7 | Biolegend | BioLegend Cat# 102016; RRID:AB_312865 | 1:400 |
| Antibody | anti-CD62L-APC | BD Biosciences | BD Biosciences Cat# 561919; RRID:AB_10895379 | 1:800 |
| Antibody | anti-CD44-PE | BD Biosciences | BD Biosciences Cat# 560569; RRID:AB_1727484 | 1:1000 |
| Antibody | anti-CD8-BV510 | Biolegend | BioLegend Cat# 100752; RRID:AB_2563057 | 1:400 |
| Antibody | anti-Thy1.2-BUV395 | BD Biosciences | BD Biosciences Cat# 565257 | 1:200 |
| Antibody | anti-Thy1.1-FITC | Biolegend | BioLegend Cat# 202504; RRID:AB_1595653 | 1:400 |
| Antibody | anti-CD19-eFlour780 | eBiosciences | Thermo Fisher Scientific Cat# 47-0193-82; RRID:AB_10853189 | 1:200 |
| Antibody | anti-CD11b-eFlour780 | eBiosciences | Thermo Fisher Scientific Cat# 47-0112-82; RRID:AB_1603193 | 1:200 |
| Antibody | anti-CD11c-eFlour780 | eBiosciences | Thermo Fisher Scientific Cat# 47-0114-80; RRID:AB_1548663 | 1:100 |
| Antibody | anti-NK1.1-eFlour780 | eBiosciences | Thermo Fisher Scientific Cat# 47–5941; RRID:AB_10853969 | 1:100 |
| Antibody | anti-CD69-FITC | Biolegend | BioLegend Cat# 104506; RRID:AB_313109 | 1:200 |
| Antibody | anti-BCL2-PE/Cy7 | Biolegend | BioLegend Cat# 633512; RRID:AB_2565247 | 1:200 |
| Antibody | anti-TNF-APC/Cy7 | BD Biosciences | BD Biosciences Cat# 560658; RRID:AB_1727577 | 1:200 |
| Antibody | anti-IFNG-Alexa647 | BD Biosciences | BD Biosciences Cat# 557735; RRID:AB_396843 | 1:1000 |
| Antibody | anti-Ki67-PE/Cy7 | Biolegend | BioLegend Cat# 652426; RRID:AB_2632694 | 1:200 |
| Antibody | anti-CD44-BUV737 | BD Biosciences | BD Biosciences Cat# 564392 | 1:400 |
| Antibody | anti-PD-1-PE/Cy7 | Biolegend | BioLegend Cat# 135216; RRID:AB_10689635 | 1:200 |
| Antibody | anti-LAG3-APC | Biolegend | BioLegend Cat# 125210; RRID:AB_10639727 | 1:200 |
| Antibody | anti-ICOS-PE | Biolegend | BioLegend Cat# 107706; RRID:AB_313335 | 1:200 |
| Antibody | anti-mouse-IL-2 (neutralizing) | Biolegend | BioLegend Cat# 503705; RRID:AB_11150768 | 10 µg/ml |

*Continued*

| Reagent type (species) or resource | Designation | Source or reference | Identifiers | Additional information |
|---|---|---|---|---|
| Antibody | anti-mouse-CD3e (stim) | Biolegend | BioLegend Cat# 100314; RRID:AB_312679 | 0.25 µg/ml |
| Antibody | anti-mouse-CD28 (co-stim) | Biolegend | BioLegend Cat# 102112; RRID:AB_312877 | 1 µg/ml |
| Other, MHC tetramer | LCMV GP33-41-specific H-2D$^b$- MHC-tetramer, PE conjugate | MBL | Cat# TS-M512-1 | 1:400 |
| Other, MHC tetramer | LCMV GP66-77-specific I-A$^b$- MHC-tetramer, APC conjugate | NIH Tetramer Core Facility | | 1:300 |
| Peptide, recombinant protein | LCMV GP33-41 peptide *KAVYNFATM* | Life Technologies | Custom synthesis | |
| Peptide, recombinant protein | LCMV GP66-77 peptide DIYKGVYQFKSV | Life Technologies | Custom synthesis | |
| Peptide, recombinant protein | Ovalbumin p257 peptide SIINFEKL | Life Technologies | Custom synthesis | |
| Peptide, recombinant protein | Ovalbumin p323 peptide ISQAVHAAHAEINEAGR | Life Technologies | Custom synthesis | |
| Peptide, recombinant protein | Recombinant mouse TNF-α | R and D Systems | Cat# 410-MT-050 | |
| Peptide, recombinant protein | Recombinant human IL-2 | Peprotech | Cat# 200–02 | |
| Peptide, recombinant protein | Recombinant mouse IL-12 | eBiosciences | Cat# 14-8121-80 | |
| Peptide, recombinant protein | Recombinant mouse IL-23 | eBiosciences | Cat# 14-8231-63 | |
| Peptide, recombinant protein | Recombinant mouse IL-6 | eBiosciences | Cat# 14–8061 | |
| Peptide, recombinant protein | Recombinant human TGF-β1 | R and D Systems | Cat# 240-B-010 | |
| Peptide, recombinant protein | T4 Polynucleotide Kinase | New England Biolabs | Cat# M0201L | |
| Peptide, recombinant protein | T4 RNA ligase 2, truncated KQ | New England Biolabs | Cat# M0373L | |
| Peptide, recombinant protein | CircLigase | Epicentre | Cat# CL4115K | |
| Peptide, recombinant protein | Phusion Polymerase | New England Biolabs | Cat# M0530L | |
| Peptide, recombinant protein | Micrococcal nuclease | New England Biolabs | Cat# M0247S | |
| Peptide, recombinant protein | RNAsin Plus | Promega | Cat# N2611 | |
| Peptide, recombinant protein | RNAse A | Affymetrix | Cat# 70194Y | |
| Peptide, recombinant protein | RNAse I | Thermo Fisher | Cat# EN0601 | |
| Peptide, recombinant protein | alkaline phosphatase | Roche | Cat# 10 713 023 001 | |
| Commercial assay or kit | Xtremegene 9 Transfection Reagent | Roche | Cat# 06 365 787 001 | |
| Commercial assay or kit | Mouse CD4 microbeads | Miltenyi | Cat# 130-049-201 | |

*Continued*

| Reagent type (species) or resource | Designation | Source or reference | Identifiers | Additional information |
|---|---|---|---|---|
| Commercial assay or kit | Mouse CD8 microbeads | Miltenyi | Cat# 130-049-401 | |
| Commercial assay or kit | Mouse CD11c microbeads | Miltenyi | Cat# 130-108-338 | |
| Commercial assay or kit | Mouse CD19 microbeads | Miltenyi | Cat# 130-052-201 | |
| Commercial assay or kit | Mouse CD11b microbeads | Miltenyi | Cat# 130-049-601 | |
| Commercial assay or kit | Mouse CD4 + CD62L + T cell isolation kit | Miltenyi | Cat# 130-093-227 | |
| Commercial assay or kit | Trizol Reagent | Life Technologies | Cat# 15596026 | |
| Commercial assay or kit | High Pure RNA Isolation Kit | Roche | Cat# 11828665001 | |
| Commercial assay or kit | Truseq RNA Library Kit | Illumina | Cat# RS-122–2001 | |
| Commercial assay or kit | Ribo-Zero rRNA removal kit | Illumina | Cat# MRZH11124 | |
| Commercial assay or kit | Cytofix/Cytoperm Kit | BD Biosciences | Cat# 554722 | |
| Commercial assay or kit | Ampure XP beads | Beckman-Coulter | Cat# A63881 | |
| Commercial assay or kit | Quant-IT dsDNA Assay Kit, High Sensitivity | Life Technologies | Cat# Q33120 | |
| Commercial assay or kit | iQ SYBR Green SuperMix | Biorad | Cat# 1708880 | |
| Commercial assay or kit | iScript cDNA Synthesis Kit | Biorad | Cat# 1708891 | |
| Chemical compound, drug | doxycycline | Sigma | Cat# D9891 | |
| Chemical compound, drug | DAPI | Sigma | Cat# 32670 | |
| Chemical compound, drug | dimethylpidilate (DMP) | Life Technologies | Cat# 21666 | |
| Chemical compound, drug | 5-bromo2'-deoxyuridine | Sigma | Cat# B9285 | |
| Chemical compound, drug | Denhardt's Solution (50X) | Life Technologies | Cat# 750018 | |
| Chemical compound, drug | cycloheximide | Sigma | Cat# C104450 | |
| Chemical compound, drug | Ribonucleoside vanadyl complexes (RVC) | New England Biolabs | Cat# S1402S | |
| Chemical compound, drug | Live-Dead Fixable Aqua | Life Technologies | Cat# L34957 | |
| Chemical compound, drug | TO-PRO-3 Iodide | Life Technologies | Cat# T3605 | |
| Software, algorithm | CLIP Toolkit (CTK) | PMID:27797762 | | |
| Software, algorithm | STAR Aligner | PMID:23104886 | | |
| Software, algorithm | Bowtie2 | PMID:22388286 | | |
| Software, algorithm | HOMER | PMID:20513432 | | |
| Software, algorithm | GenomicRanges (R Bioconductor) | PMID:23950696 | | |

*Continued on next page*

*Continued*

| Reagent type (species) or resource | Designation | Source or reference | Identifiers | Additional information |
|---|---|---|---|---|
| Software, algorithm | TxDb.Mmusculus. UCSC.mm10.ensGene (R Bioconductor) | DOI: 10.18129/B9.bioc. TxDb.Mmusculus.UCSC. mm10.knownGene | | |
| Software, algorithm | TopGO (R Bioconductor) | DOI: 10.18129/B9.bioc. topGO | | Adrian Alexa, Jorg Rahnenfuhrer |
| Software, algorithm | edgeR (R Bioconductor) | PMID:19910308 | | |
| Software, algorithm | HTseq | PMID:25260700 | | |
| Software, algorithm | Cluster 3.0 | PMID:14871861 | | |
| Software, algorithm | Java Treeview | PMID:15180930 | | |
| Software, algorithm | Gene Set Enrichment Analysis (GSEA) | PMID:16199517 | | |

## Data reporting

All data reported are for independent biological replicates, unless specifically noted in figure legends. In most cases, one mouse was one biological replicate. For CLIP studies 2–4 littermate mice of the same sex and genotype were pooled for each biological replicate. When performed, technical replicates deriving from the same biological replicate were averaged. For ex vivo studies, including genomic analyses, a sample size of 3–5 biological replicates was judged sufficient based on a power analysis using values from pilot studies, requiring $p < 0.05$ with 95% power. To account for greater variability, sample sizes were doubled for in vivo studies. Mouse studies were not blinded.

## Mice and cell maintenance

### Mice

All mouse experiments were approved by The Rockefeller University Institutional Animal Care and Use Committee regulations (Protocol 17035 hr). *Zfp36* KO mice were a generous gift from P. Blackshear (NIH; [*Taylor et al., 1996*]). BG2 mice, a TCR transgenic line specific for the class-II-restricted peptide β-gal p726 (NLSVTLPAASHAIPH) from bacterial β-galactosidase, were a generous gift from Nicolas Restifo (NIH; [*Tewalt et al., 2009*]). FoxP3-EGFP mice were obtained from Jackson Laboratories (*Haribhai et al., 2007*). Unless otherwise noted, mice of both sexes were analyzed at age 4–6, and comparisons were between littermates.

### Cell lines

293 cell lines were maintained under standard conditions (humidified, 37°C, 5% $CO_2$) in DMEM supplemented with 10% fetal bovine serum (FBS) and 0.2 mg/ml gentimicin. HEK 293T (ATCC) were authenticated by STR profiling. 293 T-rex cells were obtained directly from Life Technologies, and authenticated by functional testing of the Tet-on system. HEK cells are on the list of commonly misidentified cells lines maintained by the Internal Cell Line Authentication Committee. HEK lineage was not further confirmed here, as this was not important for these studies. J558L cells stably expressing mouse GM-CSF (a gift from Ralph Steinman; *Inaba et al., 1992*) were maintained under standard conditions in RPMI supplemented with 10% FBS, non-essential amino acids, and 0.2 mg/ml gentimicin (R10 media). Production of GM-CSF was confirmed by ELISA of culture supernatants. All cell lines were confirmed mycoplasma-free by the Bionique Testing Laboratories CellShipper service (Saranac Lake, NY).

### T cell cultures

Purified T cells were cultured with BMDCs at a 30:1 ratio and 0.2 µg/ml α-CD3 (unless otherwise noted) under standard conditions in Iscove's Minimum Defined Media (IMDM) supplemented with 10% FBS, non-essential amino acids, and 0.2 mg/ml gentimicin. Cytokine conditions, unless otherwise noted, were: Th0 (10 U/ml hIL-2); Th1 (10 U/ml hIL-2 and 5 ng/ml mIL-12); Th17 (20 ng/ml mIL-

6, 10 ng/ml mIL-23, 1 ng/ml hTGF-β); and iTreg (10 U/ml hIL-2 and 2.5 ng/ml hTGF-β). Media with fresh cytokines were replenished every 2–3 days.

## Bone Marrow-Derived Dendritic Cells
Bone marrow was flushed from tibia, femur, and pelvic bones with 26.5 gauge needles, and RBCs were lysed. Cells were plated at $10^6$ per ml in (IMDM) supplemented with 10% FBS, 0.2 mg/ml gentamycin, and a 1:20 dilution of supernatant from J558L cells stably expressing recombinant GM-CSF. Cells were refreshed on days 3, 5, and 7 with 0.5 ml additional J558L/GM-CSF supernatant. At Day 7, non-adherent cells were collected and CD11c + cells positively isolated with CD11c microbeads (Miltenyi). CD11c + cells were replated in IMDM with 1:20 J558L supernatant and 50 ng/ml recombinant mouse TNF-α (R and D Systems). At Day 9, non-adherent BMDCs were collected, washed in PBS to remove cytokines, and cryopreserved in a mix of 90% FBS and 10% DMSO.

## Experimental method details
### Pan ZFP36 antisera generation
Two pan-ZFP36 antisera (RF2046 and RF2047) were produced in rabbits by Covance against the conserved C-terminal peptide APRRLPIFNRISVSE, and successfully confirmed by ELISA, immunoblotting, and immunoprecipitation (IP). Reactivity to paralogs ZFP36, ZFP36L1, and ZFP36L2 was confirmed immunoblotting (*Figure 1—figure supplement 1A*) and IP (not shown).

### Plasmid construction
ZFP36 paralog expression plasmids were generated by PCR-amplification of coding sequences from mouse spleen cDNA, and cloning into the XhoI/NotI sites of the pOZ-N vector (*Nakatani and Ogryzko, 2003*), which has a N-terminal FLAG-HA tag and a human CD25 selection marker. Reporter plasmids were generated by cloning 3'UTR sequences downstream of an Acgfp1 cDNA. *Ifng* reporters were constructed in pcDNA3.1(+) vector, and *Tnf* and *Cd69* reporters were constructed in the doxycycline inducible pcDNA5/FRT/TO vector (both vectors from Life Technologies). The full length 3'UTR of mouse *Ifng*, *Cd69, and Tnf* and versions with the CLIP-defined ZFP36 binding sites deleted, were synthesized as gBlocks (IDT) and inserted using EcoRV and NotI sites.

### Reporter transfection assays
293 T-rex cells were maintained under standard conditions with DMEM supplemented with 10% FBS and gentamycin, in the presence of 100 ng/ml doxycycline to induce reporter expression. Cells were transfected with Xtremegene9 reagent, using a 3:1 reagent:plasmid ratio and 250 ng total DNA per well in a 24-well dish. Reporter plasmids were co-transfected with pOZ-N-mZFP36 (or empty pOZ-N vector) at a 1:1 ratio (125 ng each). 30 hr post-transfection, cells were harvested in PBS/5 mM EDTA, stained with anti-human-CD25-PE to identify pOZ-N-transfected cells, and analyzed for GFP expression by flow cytometry on the Miltenyi MACSQuant or FACSCalibur. DAPI (20 ng/ml) or TO-PRO-3 (0.1 μM; Life Technologies) were added to acquisition buffer for dead-cell exclusion. Reported MFI values are for GFP in live, CD25 +cells. RNA was harvested from replicate plates by Trizol extraction followed by cleanup, with DNAse treatment, on HiPure columns (Roche). GFP mRNA was quantified by RT-qPCR using GFP-specific primers and human GADPH primers as a reference control.

### T-cell purification
For purification of pan-CD4 +and CD8+T cells, splenocytes were cleared of RBCs by hypotonic lysis, and DC populations were depleted with CD11c microbeads (Miltenyi Biotec). T cells were then purified with CD4 or CD8 microbeads.

CD4 +naïve T cells were purified by two strategies. In most experiments, CD4 +cells were pre-enriched from pooled splenocytes and lymph nodes (LNs) by positive selection with CD4 microbeads or depletion with CD19, CD11b, and CD8 microbeads. Naïve CD4 +CD25 CD44-loCD62L-hi cells were then FACS-sorted to >99% purity. In ribosome profiling experiments and for western blot time courses (*Figure 1A*), naïve CD4 +cells were purified to >95% purity with the CD4 +CD62L + isolation kit (Miltenyi).

## T-cell treatment for RNAseq, HITS-CLIP, and ribosome profiling

For RNAseq, HITS-CLIP and ribosome profiling experiments, selective recovery of T cell RNA from T cell/DC co-cultures was achieved by using formalin-fixed DCs for co-stimulation. BMDCs were fixed prior to co-culture setup in 1% paraformaldehyde (in 1X PBS) for 5 min at room temperature, quenched with a 5-fold volume 0.4M lysine prepared in 1X PBS/5% FBS, and washed extensively in PBS/5% FBS. Pilot experiments confirmed that fixed DCs provide co-stimulatory signals to T cells and induce ZFP36 expression, though at approximately 2-fold lower efficiency than live DCs (not shown). Thus, a 15:1 T cell:DC ratio was used with fixed DCs. RT-qPCR measurements confirmed that fixation quenched recovery of DC RNA by ~100 fold (not shown), ensuring selective recovery of T cell RNA. DC-only RNAseq control samples confirmed that recovered reads from DC RNA were negligible (not shown).

## Immunoblotting

Cell lysates were prepared in lysis buffer [1X PBS/1% Igepal/0.5% sodium deoxycholate (DOC)/0.1% SDS supplemented with complete protease inhibitors and Halt phosphatase inhibitors (Roche)]. Protein concentrations were determined by Bradford assay (Biorad), and 10 µg total protein per sample were run on NuPAGE gels (Life Technologies) and transferred to fluorescence-compatible PVDF membrane (Millipore). Membranes were blocked in Odyssey PBS-based buffer (LICOR) for 1 hr to overnight, then primary antibodies were added for 1 hr at room temperature. Antibodies used for western blotting were: TTP (Sigma T5327, 1:500); ZFP36L1 (CST 2119S, 1:500); FUS (Santa Cruz sc-47711, 1:1000; or Novus NB100-562, 1:10000). After 3 washes in 1X PBS/0.05% Tween-20, membranes were incubated with fluorescent secondary antibodies (LICOR, 1:25,000) for 1 hr at room temperature. Membranes were washed 3 times in 1X PBS/0.05% Tween-20, rinsed in 1X PBS, and visualized on the Odyssey Imaging system (LICOR).

# ZFP36 HITS-CLIP

## Cell preparation and UV Cross-linking

FACS-sorted CD4 +naïve T cells were activated as described above in the presence of formalin-fixed DCs. Cells were harvested at 4 hr, or at 72 hr with 2 hr PMA/ionomycin re-stimulation. Harvested cells were UV-irradiated once at 400 mJ/cm$^2$ and once at 200 mJ/cm$^2$ in ice cold 1X PBS, pelleted, and snap-frozen until use.

## Bead preparation

ZFP36 antisera or control sera from pre-immune rabbits was conjugated to Protein A Dynabeads (Life Technologies) in binding buffer (0.1 M Na-Phosphate, pH 8.0), and washed three times to remove unbound material. IgG was covalently cross-linked to beads with 25 mM dimethylpidilate (DMP) in 0.2 M triethanolamine (pH 8.2) for 45 min at room temperature. Beads were washed twice in 0.2 M ethanolamine pH 8.0, then washed several times in PBS/0.02% Tween-20 containing 5X Denhardt's buffer. Beads were blocked overnight in the final wash prior to use.

## Lysis and immunoprecipitation

Cell pellets were re-suspended in 250 µl lysis buffer [1X PBS/1% Igepal/0.5% sodium deoxycholate (DOC)/0.1% SDS supplemented with cOmplete protease inhibitors (Roche), 10 mM ribonucleoside vanadyl complexes (RVC)]. 5 µl RQ1 DNAse (Promega) was added and incubated 5 min at 37°C with intermittent shaking. For partial digestion, NaOH was added to lysates to 50 mM and incubated at 37°C with shaking for 10 min. Alkali was neutralized by addition of equimolar HCl and HEPES pH 7.3 to 10 mM, and SuperRNAsin (Roche) was added to 0.5 U/µl. For over-digested samples, RVC was omitted from lysate preparation, and RNAse A (1:1000, USB) and RNAse I (1:100, Thermo Fisher) were added and incubated at 37°C for 5 min. Lysates were cleared by centrifugation (14,000 rpm for 10 min), and rocked with beads for 45 min to 1 hr at 4°C. Beads were washed:

- Twice lysis buffer containing 5X Denhardt's Solution
- Twice high detergent buffer (1X PBS/1% Igepal/1% DOC/0.2% SDS).
- Twice high-salt buffer (1X PBS/1% Igepal/0.5% DOC/0.1% SDS, 1M NaCl [final, including PBS])
- Three times low salt buffer (15 mM Tris pH 7.5, 5 mM EDTA)
- Twice PNK wash buffer (50 mM Tris pH 7.5, 10 mM MgCl$_2$, 0.5% Igepal)

### Post-IP on-bead processing
Alkaline phosphatase treatment, polynucleotide kinase (PNK) radiolabeling, addition of pre-adeny-lated 3' linker, SDS-PAGE, and nitrocellulose transfer and extraction were performed exactly as described (*Moore et al., 2014*).

### RNA footprint cloning
CLIP footprints were reverse-transcribed using the Br-dU incorporation and bead-capture strategy, exactly as described (*Weyn-Vanhentenryck et al., 2014*). Indexed reverse transcription (RT) primers were used (*Table 1*), allowing multiplexing of 8 samples per Miseq (Illumina) run. cDNA was circularized with CircLigase (Epicentre) as described (*Weyn-Vanhentenryck et al., 2014*), then amplified with PCR primers with Illumina sequencing adapters. Amplification was tracked with SYBR green (Life Technologies) on the iQ5 Real Time Thermocycler (Biorad), and reactions were stopped once signal reached 500–1000 relative fluorescence units (r.f.u.). Products were purified with Ampure XP beads (Beckman) and quantified with the Quant-IT kit (Life Technologies) and/or Tapestation system (Agilent). Multiplexed samples were run on the Illumina Miseq with 75 base pair single-end reads.

### RNAseq
Total RNA was extracted Trizol (Life Technologies) and further purified, with on-column DNAse treatment, using HiPure columns (Roche). 500 ng to 1 µg total input RNA was rRNA-depleted (Ribo-Zero, Illumina), and unstranded, barcoded libraries were prepared with the TruSeq RNA library kit (Illumina). Libraries were run on the Hiseq 2000, obtaining 50 bp paired-end (PE) reads.

### Ribosome profiling
#### Cell preparation
Naïve CD4 +T cells were purified with the CD4 +CD62L + isolation kit (Miltenyi) and activated in the presence of formalin-fixed DCs, as described above. After 4 hr, 100 µg/ml cycloheximide was added to cultures and briefly incubated at 37°C. Cells were harvested on ice, washed in 1X PBS containing cycloheximide, pelleted, and snap frozen in liquid nitrogen. Four pairs of biological replicates (WT and *Zfp36* KO) were analyzed.

#### Monosome and Ribosome Protected Fragment (RPF) Preparation
Cells were suspended in 0.75 ml polysome lysis buffer [20 mM HEPES pH 7.3, 150 mM NaCl, 0.5% Igepal, 5 mM MgCl$_2$, 0.5 mM DTT, complete protease inhibitors, 100 µg/ml cycloheximide, Super-RNAsin (1:1000)]. Lysates were cleared by spinning for 10 min at 2,000 rpm to pellet nuclei, followed by 10 min at 14,000 rpm for debris. For digestion of polysomes to monosomes, 1 mM CaCl$_2$ and 1500 U micrococcal nuclease (MNase, Thermo Fisher) were added to clear lysates and incubated for 45 min at room temperature with rocking. Digests were stopped with addition of 5 mM EGTA, then loaded over 10–50% w/w sucrose gradients and spun at 35,000 rpm for 3 hr. Sixteen fractions were collected using the ISCO Density Gradient Fractionation System, tracking monosome elution with UV absorbance at 254 nm.

Fractions containing monosomes and residual disomes were pooled and dialyzed against gradient buffer (20 mM HEPES pH 7.3, 150 mM NaCl, 5 mM MgCl$_2$) to remove sucrose. Ribosomes were then dissociated by adding 30 mM EDTA and 0.5 M NaCl, and freed ribosome subunits were pelleted by spinning at 75,000 rpm for 2 hr. The supernatant fraction was extracted once with acid phenol and twice with chloroform, then precipitated with standard ethanol precipitation.

#### RPF cloning
Pelleted RNA was re-suspended in water, and treated with PNK in the absence of ATP to remove 3' phosphates. RNA was re-precipitated in ethanol, and 3' linker addition was performed in 20 µl reactions containing 1 µM pre-adenylated L32 linker (*Table 1*), 400 U truncated RNA ligase 2 (NEB #M0351L), and 10% polyethylene glycol (PEG MW8000) for 2 hr at room temperature. Ligation reactions were resolved on 12.5% denaturing TBE-urea PAGE gels, and RNA fragments in the range from ~50–80 nucleotides were eluted from excised polyacrylamide fragments. RPFs were ethanol-

**Table 1.** Oligonucleotide sequences

| Primer name | Sequence | Description |
|---|---|---|
| cloning | | |
| MJM9 | ATGACTCGAGGATCTCTCTGCCATCTACGAGAGCC | mouse ZFP36 forward XhoI primer |
| MJM10 | ATGAGCGGCCGCTCACTCAGAGACAGAGATACGATTGAAGATGG | mouse ZFP36 reverse NotI primer |
| MJM266 | ATGACTCGAGACCACCACCCTCGTGTCC | mouse ZFP36L1 forward XhoI primer |
| MJM267 | ATGAGCGGCCGCTTAGTCATCTGAGATGGAGAGTCTGC G | mouse ZFP36L1 reverse NotI primer |
| MJM270 | ATGACTCGAGTCGACCACACTTCTGTCACCC | mouse ZFP36L2 forward XhoI primer |
| MJM271 | ATGAGCGGCCGCTCAGTCGTCGGAGATGGAGAGG | mouse ZFP36L2 reverse NotI primer |
| RT-qPCR | | |
| GP-f | CATTCACCTGGACTTTGTCAGACTC | LCMV RNA forward qPCR |
| GP-r | GCAACTGCTGTGTTCCCGAAAC | LCMV RNA reverse qPCR |
| MJM432 | GATTGTGGGACATCCTGGTC | mouse RPL10A forward qPCR |
| MJM433 | TCAGACCCATGACTGCTGAG | mouse RPL10A reverse qPCR |
| MJM500 | AACATCGAAGACGGCTCTGT | IFNG reporter forward qPCR |
| MJM501 | GCGCTCTGTGTGGACAAGTA | IFNG reporter reverse qPCR |
| MJM504 | CCACTACCTGAGCACCCAGT | CD69 and TNF reporters forward qPCR |
| MJM505 | GAACTCCAGCAGGACCATGT | CD69 and TNF reporters reverse qPCR |
| GAPDH-f | GTCTCCTCTGACTTCAACAGCG | human GAPDH forward qPCR |
| GAPDH-r | ACCACCCTGTTGCTGTAGCCAA | human GAPDH reverse qPCR |
| HITS-CLIP and ribosome profiling | | |
| preA-L32 | /5rApp/GTGTCAGTCACTTCCAGCGG/3ddc/ | Pre-Adenylated 3' ligation linker |
| RT1 | /5Phos/DDDCGATNNNNNNNNAGATCGGAAGAGCGTCGT/iSp18/CACTCA/iSp18/**CCGCTGGAAGTGACTGAC** | Indexed RT primer |
| RT2 | /5Phos/DDDTAGCNNNNNNNNAGATCGGAAGAGCGTCGT/iSp18/CACTCA/iSp18/**CCGCTGGAAGTGACTGAC** | Indexed RT primer |
| RT3 | /5Phos/DDDATCGNNNNNNNNAGATCGGAAGAGCGTCGT/iSp18/CACTCA/iSp18/**CCGCTGGAAGTGACTGAC** | Indexed RT primer |
| RT4 | /5Phos/DDDGCTANNNNNNNNAGATCGGAAGAGCGTCGT/iSp18/CACTCA/iSp18/**CCGCTGGAAGTGACTGAC** | Indexed RT primer |
| RT5 | /5Phos/DDDCTAGNNNNNNNNAGATCGGAAGAGCGTCGT/iSp18/CACTCA/iSp18/**CCGCTGGAAGTGACTGAC** | Indexed RT primer |
| RT6 | /5Phos/DDDGATCNNNNNNNNAGATCGGAAGAGCGTCGT/iSp18/CACTCA/iSp18/**CCGCTGGAAGTGACTGAC** | Indexed RT primer |
| RT7 | /5Phos/DDDAGTCNNNNNNNNAGATCGGAAGAGCGTCGT/iSp18/CACTCA/iSp18/**CCGCTGGAAGTGACTGAC** | Indexed RT primer |
| RT8 | /5Phos/DDDTCGANNNNNNNNAGATCGGAAGAGCGTCGT/iSp18/CACTCA/iSp18/**CCGCTGGAAGTGACTGAC** | Indexed RT primer |
| DP5-PE | AATGATACGGCGACCACCGAGATCTACACTCTTTCCCTACACGACGCTCTTCCGATCT | forward PCR primer, Illumina adapter |

*Table 1 continued*

| Primer name | Sequence | Description |
| --- | --- | --- |
| SP3-PE | CAAGCAGAAGACGGCATACGAGATCTCGGCATTCCTGCCGCTGGAAGTGACTGACAC | reverse PCR primer, Illumina adapter |
| PE-R1 | ACACTCTTTCCCTACACGACGCTCTTCCGATCT | sequencing primer (standard Illumina read 1) |

DOI: https://doi.org/10.7554/eLife.33057.022

precipitated and cloned using the exact Br-dU incorporation and bead-capture method used for HITS-CLIP (see above) and described elsewhere (*Weyn-Vanhentenryck et al., 2014*).

## Flow cytometry
### Surface and intracellular stains
Antibodies used for flow cytometry are listed in the Key Resource Table. Flow cytometry data were acquired on the BD LSR Fortessa or LSR II systems. Cell sorting was done on the BD FACSAria. For surface stains, cells incubated with antibodies in FACS buffer (1X PBS/1% FBS) for 20 min at 4°C, then washed twice in FACS buffer. For live cell analysis, samples were then re-suspended in FACS buffer containing 20 ng/ml DAPI and acquired directly. For fixed samples, cells were washed twice with PBS after surface staining, then incubate with Live/Dead Fixable Aqua (1:1000 in PBS) for 10 min at RT. Cells were washed once with FACS buffer, then fixed and permeabilized using the BD Cytofix/CytoPerm kit. For intracellular stains, cells were then incubated with antibodies diluted in 1X Perm/Wash buffer for 30 min at RT, washed twice with Perm/Wash buffer, and acquired.

Unless noted otherwise, flow cytometry data were gated on live (DAPI- or Aqua-negative) single cells, with additional marker gates applied as indicated.

### Intracellular cytokine staining (ICS)
For ICS, cells were stimulated with PMA (20 ng/ml) and ionomycin (1 µM) for 5–6 hr in the presence of GolgiStop (BD) prior to harvesting. Cells were processed and analyzed as described above.

Cytokine production in LCMV studies was measured in response to class-I H-2$^d$-restricted GP33-41 (KAVYNFATM) and class-II I-A$^b$-restricted GP66-77 (DIYKGVYQFKSV) LCMV peptides. $2 \times 10^6$ splenocytes were incubated in 1 ml R10 media (RPMI with 10% FBS, 50 µM β−ME, non-essential amino acids, and gentamycin) with GolgiStop and 0.2 µg/ml class-I peptides or 2 µg/ml class-II peptides. After 5–6 hr, cells were harvested and processed as described above. Gates were established using splenocytes from naïve mice and LCMV-infected mice pulsed with irrelevant peptides (class-I OVA p257 [SIINFEKL]; class-II OVA p323 [ISQAVHAAHAEINEAGR]).

### MHC tetramers
LCMV GP33-41-specific H-2D$^b$-restricted MHC-tetramer was purchased as a PE conjugate from MBL. APC-conjugated LCMV GP66-77-specific I-A$^b$-restricted MHC-tetramer was a kind gift from the NIH Tetramer Core Facility. For class I tetramer, staining was done as described for surface antibody staining using a 1:800 dilution in FACS buffer. For class II tetramer, staining was done at a 1:300 dilution at RT for 1 hr, then cells were washed once in FACS buffer before staining with surface antibodies as described above. In all experiments, tetramer +cell gates were established using naïve animals and irrelevant, class-matched tetramers as negative controls.

## Thymidine incorporation assays
200 µl T cell-DC co-cultures were set up in 96-well plates, and 1 µCi $^3$H-thymidine was added at indicated time points. Cultures were harvested onto glass filter plates (Perkin-Elmer) and dried thoroughly, before addition of 30 µl scintillation fluid per well and acquisition on the Topcount scintillation counter (Perkin-Elmer).

## Apoptosis assays

24 hr after activation, cultures were harvested on ice and stained with surface markers. Prior to acquisition, buffer was removed from samples and Annexin-V-staining buffer containing Annexin-V-PE (BD Biosciences, 2 µl per samples) and 20 ng/ml DAPI was added.

## Bone Marrow Chimeras

Host CD45.1 mice (Jackson Laboratory 002014) were given a lethal dose of gamma radiation (900 rads), then intravenously injected with a 1:1 mix of Thy1.1 WT and Thy1.2 *Zfp36* KO BM cells ($4-5 \times 10^6$ total cells). Chimeras were analyzed 10–12 weeks after re-constitution.

## LCMV studies

Mice were infected with LCMV Armstrong strain by intraperitoneal injection of $1 \times 10^5$ plaque forming units. At indicated time points, peripheral blood was collected by retro-orbital bleeding and analyzed for virus-specific T cells using MHC-tetramers (see above). At 6, 10, and 40 days p.i., spleens were analyzed for LCMV-specific T cells with MHC-tetramers and analysis of cytokine production in response to LCMV peptide antigens (see above). LCMV genomic RNA was quantified in total spleen RNA by RT-qPCR. To determine copy number, a standard curve was generated using a gBlock comprising the LCMV PCR amplicon. LCMV copy number per spleen was calculated using a linear regression of the standard curve, and scaling by appropriate dilution factors.

## Quantification and statistical analysis

Details for statistical analysis appear in figure legends. Comparisons between experimental groups were done with two-sided student's t-tests, with $p<0.05$ considered significant. Statistics for bioinformatic analyses are detailed below.

## Bioinformatics

### HITS-CLIP

Processing and alignment of HITS-CLIP read data for Br-dU CLIP was done as described ([*Shah et al., 2017*; *Moore et al., 2014*; *Weyn-Vanhentenryck et al., 2014*]). Peak calling was done with the CLIP Toolkit (CTK) command tag2peak.pl, requiring enrichment over background with FDR < 0.01 (*Shah et al., 2017*). For background determination, a genic model was used including known mouse transcripts extended 10 kb downstream to include non-annotated 3'UTR variants. For analyses in *Figure 1E–G* and *Figure 1—figure supplement 2C*, peaks were defined and analyzed separately for 5 WT (*Supplementary file 1A*) and 3 *Zfp36* KO biological replicates (*Supplementary file 1B*). In these analysis, bindings sites were defined as peak height [PH]>5, from ≥3 biological replicates, with two different antisera. Subsequently, the 5 WT and 3 KO replicates (eight total) were pooled for peak definition to increase depth and sensitivity (*Supplementary file 1D*). For these analysis, peaks had to be supported by two different antisera and >5 biological replicates, ensuring support from both WT and KO datasets. Cross-link-induced truncations (CITS) were also determined as described, pooling all biological replicates from all time points ([*Shah et al., 2017*]; *Supplementary file 1D*).

Motif enrichment analysis was done on a 30 nucleotide window surrounding peaks supported in at least three biological replicates in the indicated dataset. HOMER (*Heinz et al., 2010*) was used with commands similar to: findMotifsGenome.pl peak_file.txt mm10 output_folder -rna -len 8

Site annotation was done with custom R scripts using the GenomicRanges and TxDb.Mmusculus. UCSC.mm10.ensGene (Transcript Database for mm10).

T cell activation time course data were downloaded from GEO, and genes were clustered by expression patterns using k means partitioning in Cluster 3.0 (*de Hoon et al., 2004*). An optimal k of 20, as previously determined (*Yosef et al., 2013*), was re-confirmed by visualizing a wide range of k values in Java Treeview (*Saldanha, 2004*). The distribution of ZFP36 3'UTR and CDS CLIP targets among k clustered was evaluated with Fisher's exact test. Average expression values of genes in the three most enriched and depleted clusters were plotted. Gene Ontology enrichments were determined with the TopGO package in R (*Alexa and Rahnenfuhrer, 2016*).

## RNAseq

RNAseq reads were aligned against the mouse reference genome (mm10) with STAR with default settings (*Dobin et al., 2013*). Feature counts per gene were obtained with HTSeq (*Anders et al., 2015*). Differential gene expression was analyzed with edgeR using a standard t-test pipeline (*Robinson et al., 2010*). For CDF analysis (e.g. *Figure 1H*), $\log_2$-fold-change values derived from edgeR were plotted. Analysis included genes with average RPKM >3 in WT or *Zfp36* KO biological replicates in RNAseq, and with sufficient coverage in ribosome profiling (see below) to permit quantification. Genes were classified based on the presence of robust ZFP36 HITS-CLIP peaks (FDR < 0.01, supported by >5 biological replicates) in 3'UTR or CDS. Non-exclusive site annotation was used here, meaning some genes have both 3'UTR and CDS sites. Analysis was repeated with exclusive annotation, with similar results (not shown). The negative control set was defined as genes without significant ZFP36 binding in any transcript region. Differences between gene sets were evaluated with a two-tailed Kolmogorov-Smirnov test.

The gene expression profile observed 72 hr after activation of *Zfp36* KO cells was analyzed for significantly overlapping gene sets using GSEA and the Molecular Signature Database (*Subramanian et al., 2005*). Overlap with published profiles of CD4 +T cell exhaustion (*Crawford et al., 2014*) was determined with a GSEA pre-ranked analysis.

## Ribosome profiling

Initial filtering and processing of ribosome profiling reads was done as for HITS-CLIP. Reads were aligned against a mouse reference transcriptome (Ensembl) using Bowtie 2, and read counts per transcript were calculated (*Langmead and Salzberg, 2012*). Differential expression analysis was done with edgeR using a paired general linear model (GLM). Only genes with cpm (counts per million read) values > 1 in at least two biological replicates of any genotype were included. CDF analyses were done as described for RNAseq. The change in translation efficiency (ΔTE) between *Zfp36* KO and WT cells was calculated for each mRNA as the difference in log2-fold-change values (KO/WT) from ribosome profiling and RNAseq. mRNAs were ranked on the ΔTE metric, and distribution of ZFP36 CLIP targets therein was evaluated with a pre-ranked GSEA analysis.

Ribosome profiling read coverage from pooled biological replicates of each genotype was calculated across individual mRNAs with a sliding 20 nucleotide window, normalizing for dataset read depth. Differences between WT and *Zfp36* KO coverage were evaluated with a binomial test. For the metagene coverage plot, all transcripts with at least 10 reads in the region of interest were included and the proportion of reads which centered at each nucleotide position was calculated. The mean coverage for each position is shown relative to the central coding region with additional normalization included for the number of transcripts represented at each position.

## Data availability

High throughput sequencing data from this study are available at the NCBI Gene Expression Omnibus (GEO) website under accession GSE96076.

## Acknowledgements

We thank Yuan Yuan and other Darnell lab members for their support and insightful comments. We are indebted to Svetlana Mazel and the Rockefeller FCRC staff for cell sorting and flow cytometry expertise. MJM. was supported by a fellowship from the Jane Coffin Childs Fund for Medical Research. This work was supported by grants to RBD. from the National Institutes of Health (NS034389, NS081706, R35NS097404), and the Starr Cancer Consortium. RBD. is an Investigator of the Howard Hughes Medical Institute.

## Additional information

**Competing interests**

Michael J Moore: currently affiliated with Regeneron Phrmaceuticals. The author has no other financial competing interests to declare. The other authors declare that no competing interests exist.

## Funding

| Funder | Grant reference number | Author |
| --- | --- | --- |
| Jane Coffin Childs Memorial Fund for Medical Research | | Michael J Moore |
| National Institutes of Health | NS034389 | Robert B Darnell |
| Starr Foundation | | Robert B Darnell |
| National Institutes of Health | NS081706 | Robert B Darnell |
| National Institutes of Health | R35NS097404 | Robert B Darnell |

The funders had no role in study design, data collection and interpretation, or the decision to submit the work for publication.

## Author contributions

Michael J Moore, Conceptualization, Data curation, Formal analysis, Supervision, Funding acquisition, Validation, Investigation, Visualization, Methodology, Writing—original draft, Project administration, Writing—review and editing; Nathalie E Blachere, Conceptualization, Formal analysis, Investigation, Methodology, Project administration, Writing—review and editing; John J Fak, Validation, Investigation, Methodology; Christopher Y Park, Data curation, Formal analysis, Methodology; Kirsty Sawicka, Software, Formal analysis, Investigation, Methodology; Salina Parveen, Ilana Zucker-Scharff, Investigation, Methodology; Bruno Moltedo, Resources, Investigation, Methodology; Alexander Y Rudensky, Resources, Investigation, Methodology, Writing—review and editing; Robert B Darnell, Conceptualization, Resources, Supervision, Funding acquisition, Investigation, Methodology, Writing—original draft, Project administration, Writing—review and editing

## Author ORCIDs

Michael J Moore (iD) http://orcid.org/0000-0003-2358-083X
Robert B Darnell (iD) https://orcid.org/0000-0002-5134-8088

## Ethics

All mouse experiments were approved by The Rockefeller University Institutional Animal Care and Use Committee regulations (Protocol 17035-H).

## Decision letter and Author response

Decision letter https://doi.org/10.7554/eLife.33057.034
Author response https://doi.org/10.7554/eLife.33057.035

# Additional files

## Supplementary files

• Supplementary file 1. ZFP36 binding sites in CD4 +T cells 4 hr post-activation (attached spreadsheet). Pan-ZFP36 HITS-CLIP peaks (A) in WT CD4 +T cells 4 hr post-activation (B) in *Zfp36* KO CD4 +T cells 4 hr post-activation. (C) identified only in *Zfp36* KO and not WT cells, and (D) pooled from WT and *Zfp36* KO samples. (E) Cross-link induced truncation (CITS) cites from all pooled ZFP36 HITS-CLIP data (FDR < 0.01).
DOI: https://doi.org/10.7554/eLife.33057.023

• Supplementary File 2. Gene Ontology enrichments for ZFP36 target mRNAs in CD4 +T cells, 4 hr post-activation (attached spreadsheet).
DOI: https://doi.org/10.7554/eLife.33057.024

• Supplementary File 3. ZFP36 binding sites in CD4 +T cells 72 hr post-activation (attached spreadsheet).
DOI: https://doi.org/10.7554/eLife.33057.025

• Transparent reporting form
DOI: https://doi.org/10.7554/eLife.33057.026

## Data availability

Sequencing data are in GEO under the accession code GSE96076

The following dataset was generated:

| Author(s) | Year | Dataset title | Dataset URL | Database, license, and accessibility information |
|---|---|---|---|---|
| Robert B Darnell | 2018 | ZFP36 RNA-binding proteins restrain T-cell activation and anti-viral immunity | https://www.ncbi.nlm.nih.gov/geo/query/acc.cgi?acc=GSE96076 | Publicly available at the NCBI Gene Expression Omnibus (accession no:GSE96076) |

The following previously published dataset was used:

| Author(s) | Year | Dataset title | Dataset URL | Database, license, and accessibility information |
|---|---|---|---|---|
| Nir Yosef | 2013 | Reconstruction of the dynamic regulatory network that controls Th17 cell differentiation by systematic perturbation in primary cells | https://www.ncbi.nlm.nih.gov/geo/query/acc.cgi?acc=GSE43970 | Publicly available at the NCBI Gene Expression Omnibus (accession no: GSE43955) |

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
