## [Decision Letter]

Thank you for submitting your article "ZFP36 RNA-binding proteins restrain T-cell activation and anti-viral immunity" for consideration by *eLife*. Your article has been favorably evaluated by James Manley (Senior Editor) and three reviewers, one of whom, Douglas Black, is a member of our Board of Reviewing Editors. The reviewers have opted to remain anonymous.

The reviewers have discussed the reviews with one another and the Reviewing Editor has drafted this decision to help you prepare a revised submission.

Summary:

Moore et al. examine the role of the ZFP36 proteins in T cell activation and immune function. This family of proteins, including *Zfp36* (also known as TTP), *Zfp36L1*, and *Zfp36L2*, are well known mediators of mRNA instability through binding to AU-rich elements (ARE). Immune cytokines such as TNF and IFNg are among their targets. The phenotype of *Zfp36* loss in mice has mainly been studied in myeloid cells but it is expected that these proteins play broader roles in immune cell function. Here the authors show that the *Zfp36* and *Zfp36L1* proteins are strongly induced by T cell activation. Using an antibody cross-reactive with all the paralogs, they perform CLIP to identify a large set of RNA binding sites across the transcriptome. They compare the binding sites of both proteins identified in WT mice with those of presumably *Zfp36L1* alone in *Zfp36* KO mice. These datasets show broad overlap, yield essentially identical binding motifs with the expected ARE, and thus the two proteins apparently bind largely the same target RNAs. The authors show that transcripts containing CLIP peaks in their 3'UTR or CDS exhibit increased expression in the KO mice, as expected if the proteins mediate translational repression or targeted mRNA decay. Several transcripts exhibiting substantial binding of *Zfp36* in the 3'UTR, including *Cd69* and *Tnf*, show minimal change in RNA levels in the KO but a significant increase in protein, implying an effect on translation. This is recapitulated in a reporter assay using the IFNg 3'UTR. RNA expression from the reporter does go down in response to co-expressed *Zfp36*, but the protein product shows a larger effect. They performed ribosome profiling in WT and KO mice and correlate this with RNAseq to determine translational efficiencies. They find that transcripts with *Zfp36* binding in the 3'UTR and more strongly with binding in the CDS exhibit increased ribosome association and translational efficiency in the KO cells. The *Zfp36* target set includes mRNAs for many types of proteins but is enriched for transcripts affecting cellular processes such as cell proliferation, apoptosis, T cell activation and others. Comparing WT and KO T cells, the authors show that the loss of *Zfp36* leads to reduced apoptosis and increased proliferation after stimulation. Using a transgenic carrying a β-Gal specific TCR, they show that these results hold true for true T cell activation by antigen, with the activated KO cells showing enhanced CD69 and CD25 induction. Thus, the loss of *Zfp36* leads to enhanced T cell activation. Generating mixed WT and KO bone marrow chimeras, they show that sorted KO cells still exhibit enhanced proliferation, but if the sorted cells are mixed back with WT this effect was reduced. It thus appears that the enhanced T cell activation is autonomous to the mutant cells but can be affected by autocrine factors if WT cells are present. Repeating some of their analyses at 3 days post stimulation rather than 4 hours, they found larger numbers and magnitudes of expression changes in the KO cells. These were not correlated with *Zfp36* binding, indicating they were indirect downstream effects, and overlapped with known signatures of T cell exhaustion. Looking specifically at proteins associated with exhaustion PD-1, ICOS, and LAG-3, they confirmed their increased expression and found them to be dependent on Th1 cytokines. Finally, they challenged the WT and KO mice with LCMV infection. They found that the KO mice mounted a more rapid response to virus, with increased CD4 and CD8 positive cells in both blood and spleen, increased IFN and TNF production, and decreased LCMV genomic RNA in the spleen early after infection.

There is a lot of data here and all the reviewers found the genomewide RNA binding and translation analyses for these proteins to be well done and useful. Although post-transcriptional control of cytokine production has been studied for many years, there is much to be learned regarding both its mechanisms and biology. The observation that *Zfp36* may be altering translation as well as mRNA decay is novel, but its underlying mechanism is not pursued. How the *Zfp36* regulatory program fits into the context of acquired immunity is also not understood, and in showing that *Zfp36* provides a brake on T cell responses, this study adds potential new understanding in this area. However, the loss of *Zfp36* alone, without the *Zfp36L1* KO, yielded often small or subtle effects. Without some assessment of the consequences of depleting both proteins, additional characterization is needed to solidify the conclusions related to the role of these proteins in T cell biology.

Essential revisions:

1) LCMV infection should be performed in mixed chimeras to determine whether the degree to which the observed effects on the kinetics of the T cell response are cell intrinsic.

2) One important unaddressed issue is the mechanism driving the increase in ZNF36 expression upon CD4^+^ activation. In the absence of a careful analysis of this, the authors should at least discuss what the RNA-Seq, HITS-CLIP and ribosome profiling suggests.

3) The authors make the assumption that the heterogeneity in Figure 1A and other western blots of ZNF36 is due to differential phosphorylation. However, the blots of individual paralogs in Figure 1—figure supplement 1 suggest that these proteins run as discrete bands. It would be interesting to see the gel in Figure 1A blotted with the paralog-specific antibodies to determine if the multiple species are a result of the distinct isoforms. This is of interest as the detection of the individual species changes over time (faster migrating species dominates in the first 1-2 hours, followed by the slower migrating species).

4) Figure 1H and many of the protein blots in Figure 2 show statistically significant differences, but are these really biologically meaningful? It seems particularly unclear that the 10% difference in IFN-γ protein in Figure 2A could give rise to the large differences in IFN-γ secretion in Figures 6 and 7. Is the latter perhaps due to an indirect effect of increased Th1 polarization? (see point 7),

5) The proposed role of ZNF36 in repressing translation is not well supported. The results of the reporter assay in Figures 2B and C are striking, but the effect on protein is much greater than observed in the endogenous gene (Figure 2A). This may not be due to binding of ZNF36 to the UTR since mutation of the binding site has an effect even in the absence of ZNF36. (Alternatively, 293 cells may express ZNF36 – a western blot should be done.) Or is this due to some other feature of the IFN-γ UTR? What is the result if the TNF or CD69 3' UTR is used? Since these endogenous genes seem to show a larger response, they may give more robust results with the reporter assay.

6) A second concern with the translation conclusion is it is unclear in Figure 3C why "intron bound" is used as the negative control rather than "none". Transcripts that are enriched for binding in the intron may well produce mRNAs whose expression or function differs between WT and KO, confounding the interpretation of the translation differences between conditions (if that is what the GSEA score reveals – this is unclear).

7) The authors should show what percentage of cells in the in vitro activation in Figures 1 and 6 are actually polarized to a Th1 fate. Typically only a subset of the cells actually achieve the desired fate under such in vitro conditions. Given that T cells in the ZNF36 KO mice appear predisposed to a Th1 fate, it is important to assess whether the differences observed in Figure 6 are a direct effect of ZNF36 or simply reflect that a greater percentage of cells in the activation culture are Th1 cells.

8) It would be much better to use FoxP3 instead of CD25 to identify Tregs (e.g. in Figure 7—figure supplement 1F).

---

## [Author Response]

Essential revisions:1) LCMV infection should be performed in mixed chimeras to determine whether the degree to which the observed effects on the kinetics of the T cell response are cell intrinsic.

These analyses were done for this revision, and appear in Figure 7—figure supplement 3. Quoting from the revised Results:

“To examine whether the accelerated LCMV-specific T-cell response in *Zfp36* KO mice is cell-intrinsic, infections were repeated in mixed BM chimeras. […] Of note, maximum T-cell expansion was observed in mixed chimeras 7-8 days p.i., which was intermediate to the maxima observed in *Zfp36* KO (6 days) and WT (10 days) mice.”

And the revised Discussion:

“In mixed BM chimeras, the response kinetics of WT and *Zfp36* KO T cells to LCMV infection were indistinguishable. […] Regardless of the initiating mechanism, the observation of enhanced LCMV clearance is likely T cell-dependent, given the central role of T cells in LCMV immunity (Matloubian et al., 1994).”

2) One important unaddressed issue is the mechanism driving the increase in ZNF36 expression upon CD4^+^ activation. In the absence of a careful analysis of this, the authors should at least discuss what the RNA-Seq, HITS-CLIP and ribosome profiling suggests.

We have included additional data from analyses of naïve CD4^+^ T cell RNAseq data that address this point. Quoting from the revised Results:

“The characterization of *Zfp36* as an immediate early response gene in various cell types established transcription as a mechanism of its activation-induced expression (Lai et al., 1995). […] These observations indicate that post-transcriptional mechanisms regulate expression of ZFP36 paralogs in T cells.”

And quoting from the revised Discussion:

“ZFP36 and ZFP36L1 expression are rapidly induced upon T cell activation, and gradually recede thereafter. [...] However, the presence of its mRNA further suggests post-transcriptional control of ZFP36 paralog expression, and is consistent with functions in other contexts or stages of T cell function ((Vogel et al., 2016; Hodson et al., 2010)).”

3) The authors make the assumption that the heterogeneity in Figure 1A and other western blots of ZNF36 is due to differential phosphorylation. However, the blots of individual paralogs in Figure 1—figure supplement 1 suggest that these proteins run as discrete bands. It would be interesting to see the gel in Figure 1A blotted with the paralog-specific antibodies to determine if the multiple species are a result of the distinct isoforms. This is of interest as the detection of the individual species changes over time (faster migrating species dominates in the first 1-2 hours, followed by the slower migrating species).

We agree with the reviewers’ interpretation – the signal in Figure 1A reflects detection of two paralogs – ZFP36 and ZFP36L1 – and the balance may be shifting over time. In addition, the observed bands running well above the expected MW of ZFP36 (36 kD) are strongly consistent with previously described, hyperphosphorylated species. We have clarified and expanded our thoughts on this heterogeneity in the Results (subsection “ZFP36 dynamics during T cell activation”, second paragraph).

Regarding a deeper analysis of the dynamics of individual paralogs, we can offer further insights for the reviewers’ benefit. At the time of analysis, the blot in Figure 1A was stripped and re-probed with the ZFP36L1-specific antibody (this antibody also reacts with the significantly larger ZFP36L2, but none was detected).

Please note the diminished quality is due to the harsh conditions that were necessary for complete stripping, which was confirmed by re-probing with secondary antibody. Nonetheless, the pattern here is clearly similar to that of the pan-ZFP36 antisera, with the higher band dominating initially, and the lower band increasing over time. As in plots with pan ZFP36 sera, the higher band in these blots runs significantly above the expected MW of ZFP36/L1 (~36 kD). Since only a single paralog is represented in the re-probed blot, we interpret the higher band as hyperphosphorylated, based on its consistency with numerous prior descriptions of ZFP36 paralog phosphorylation (Ross et al., 2015; Qiu et al., 2012; Sun et al., 2006).

As the reviewers note, a ‘higher’ band is the only one detected in blots with the ZFP36-specific antibody 4 hours after activation in Figure 1—figure supplement 1D. A working ZFP36-specific antibody was not identified until a year after these time courses were done, after screening many commercial products. This antibody gave a relatively low signal, even at high titers, compared to the pan-ZFP36 and ZFP36L1/2 antibodies. While sufficient to confirm loss of ZFP36 in KO cells in Figure 2—figure supplement 1D, this antibody is not suitable for a conclusive probe of paralog isoform kinetics. However, given the high homology between ZFP36 and ZFP36L1 and their similar predicted size, it is very likely the band in question (Figure 1—figure supplement 1D, ZFP36 panel) is hyperphosphorylated ZFP36, and that hypophosphorylated ZFP36 is below detection.

It is further worth noting that published RNA expression data that showed similar expression dynamics for ZFP36 and ZFP36L1 after T cell activation (for instance, the time course data from Yosef et al., 2013 – Author response image 2).

**Author response image 2. respfig2:** 

Taken together, existing evidence suggests ZFP36 and ZFP36L1 show similar patterns of expression after T cell activation. We have not included these details in the revised manuscript, because we are not aware of the necessary reagents for a full analysis and, more importantly, we lack evidence of distinct paralog functions in this context. As stated in the manuscript (Results subsection “Transcriptome-wide identification of ZFP36 target RNAs in CD4+ T cells”, third paragraph), future studies may tease out differences in paralog function, but our data suggest are large degree of redundancy.

4) Figure 1H and many of the protein blots in Figure 2 show statistically significant differences, but are these really biologically meaningful? It seems particularly unclear that the 10% difference in IFN-γ protein in Figure 2A could give rise to the large differences in IFN-γ secretion in Figures 6 and 7. Is the latter perhaps due to an indirect effect of increased Th1 polarization? (see point 7).

The first critical point to consider is that all of the genomic and functional analyses begin with purified, naïve T cells, and we have shown that naïve ZFP36 KO and WT CD4^+^ T cells differentiate similarly to Th1, Th17, and Treg lineages (Figure 7—figure supplement 1G, I). Therefore, the observed differences in protein levels and ribosome profiling cannot be attributed to differential Th1 skewing, and we have no evidence that loss of ZFP36 intrinsically disposes naïve T cells to a specific fate, Th1 or otherwise.

There may be understandable confusion caused by the data in Figure 7—figure supplement 1H, which shows that splenocytes stimulated directly ex vivo with PMA/ionomycin (no skewing) contain more IFNG-producing T cells in ZFP36 KO versus WT, although only marginally for CD4^+^ cells. This data was included as part of a descriptive analysis of the in vivo T-cell compartment of ZFP36 KO mice. The splenic cells producing high levels of IFNG directly ex vivo are effector and/or memory cells, and are not the population analyzed in our other experiments. This result does indicate that Th1 effector cells are more abundant in ZFP36 KO versus WT animals at the age analyzed (6 weeks). However, this accumulation probably occurs over time, and we cannot conclude it is T cell-intrinsic. Again, when naïve T cells are sorted, and analyses are conducted in a short time window, there is no detectable difference in the ability of WT and KO cells to form Th1 cells.

The second key point to consider is that genomic and functional analyses were done at an early time point (4 hours) well before full skewing has occurred. At this point, as expected and as the reviewers note, IFNG protein expression is quite low. Full differentiation to naïve cells to Th1 cells under these conditions, with high levels of IFNG expression, takes 48-72 hours. Nonetheless, analysis of this early time point was crucial because it represents peak ZFP36 expression (Figure 1) and because we have observed that likely secondary effects pre-dominate later on (Figure 6). These analyses were designed to most effectively observe the regulatory effects of ZFP36, not the maximum biological impact of any specific target, including IFNG. That said, we agree the mentioned 10% difference does not fully account for the larger ones observed in secreted levels of IFNG in Figure 6. Crucially, the latter experiment measured secreted IFNG accumulated in culture supernatants over 3 days that are also impacted by the greater expansion of ZFP36 KO cells under these conditions, due to more rapid activation (see Results subsection “Downstream effects of ZFP36 regulation”). The compounding effect of small differences over time, combined with greater cell numbers, can readily account for the larger differences at the later time point.

Overall, these points highlight two critical themes – 1) that small effects at early time points can have a large impact in a rapidly changing and expanding cell population, such as differentiating T cells; and 2) that subtle effects on many targets, rather than large effects on individual ones, are the norm rather than the exception in post-transcriptional control. We have inserted comments in the Discussion (seventh paragraph) to make these points more effectively.

5) The proposed role of ZNF36 in repressing translation is not well supported. The results of the reporter assay in Figures 2B and C are striking, but the effect on protein is much greater than observed in the endogenous gene (Figure 2A). This may not be due to binding of ZNF36 to the UTR since mutation of the binding site has an effect even in the absence of ZNF36. (Alternatively, 293 cells may express ZNF36 – a western blot should be done.) Or is this due to some other feature of the IFN-γ UTR? What is the result if the TNF or CD69 3' UTR is used? Since these endogenous genes seem to show a larger response, they may give more robust results with the reporter assay.

For this revision we have generated and tested new reporters based on TNF and CD69 3’UTRs, which show results very similar to the IFNG reporter (Figure 2C). The key result from these assays is that effects on protein output are significantly greater than on RNA levels, which supports the principle of translation regulation by ZFP36. Indeed the CD69 and TNF reporters show a more pronounced effect than IFNG in this regard.

As noted by the reviewers, deleting the ZFP36 binding site had an effect on GFP protein expression both in the presence and absence of transfected ZFP36. This was also observed with the TNF and CD69 reporters, indicating it is not a unique feature of the IFNG 3’UTR. Western blots in Figure 1—figure supplement 1A rule out the endogenous expression of ZFP36 in 293 cells as a confounding factor. Rather, this effect probably reflects the activity of other ARE regulatory factors in 293 cells. Nonetheless, ZFP36-dependent effects on reporter expression are clear – we have added additional labels with fold-changes to Figure 2C and re-worked the revised results to aid interpretation.

6) A second concern with the translation conclusion is it is unclear in Figure 3C why "intron bound" is used as the negative control rather than "none". Transcripts that are enriched for binding in the intron may well produce mRNAs whose expression or function differs between WT and KO, confounding the interpretation of the translation differences between conditions (if that is what the GSEA score reveals – this is unclear).

This is the comparison that was done, and we have clarified this point with a more detailed explanation in the revised Results (subsection “ZFP36 represses target mRNA abundance and translation during T cell activation”, last paragraph). The GSEA score is determined by ranking all observed transcripts by the metric of interest (in this case, translational efficiency), and examining the distribution of a designated subset (in this case, ZFP36 CLIP targets) in the whole set. The requested comparison (“target” vs. “none”) is built into this design, and is the basis of the reported p-values. We included the “intron bound” set as an example of a rationally defined subset that did not show such an enrichment. This supports the specific function of CDS and 3’UTR sites in regulating translation.

7) The authors should show what percentage of cells in the in vitro activation in Figures 1 and 6 are actually polarized to a Th1 fate. Typically only a subset of the cells actually achieve the desired fate under such in vitro conditions. Given that T cells in the ZNF36 KO mice appear predisposed to a Th1 fate, it is important to assess whether the differences observed in Figure 6 are a direct effect of ZNF36 or simply reflect that a greater percentage of cells in the activation culture are Th1 cells.

Addressed above with point 4). Skewing percentages are shown in Figure 7—figure supplement 1I.

8) It would be much better to use FoxP3 instead of CD25 to identify Tregs (e.g. in Figure 7—figure supplement 1F).

We acknowledge the point that FoxP3 is the definitive marker for Tregs. We did not examine FoxP3 expression in splenic T cells directly ex vivo, but feel that a major difference in Treg presence in vivo would have been apparent in the enumeration of CD25-hi CD4^+^ cells, which is a widely used correlate. We did examine the capacity for iTreg induction in vitro in WT and KO FoxP3-GFP mice, and observed no significant difference (Figure 7—figure supplement 1G).

To address this concern, we have re-worded the Results and accompanying figure labels:

“Levels of CD25-hi CD4^+^ cells were not significantly different in spleens of WT and KO mice, consistent with similar levels of natural Tregs (Figure 7—figure supplement 1F). FoxP3 expression was not examined directly ex vivo, but in vitro induction of Tregs from naïve cells CD4^+^ T-cells, enumerated in FoxP3-GFP mice, was not different between WT and KO (Figure 7—figure supplement 1F).”